# Small molecule inhibition of Dynamin-dependent endocytosis targets multiple niche signals and impairs leukemia stem cells

Cedric S. Tremblay [ID] [1✉], Sung Kai Chiu[1,2], Jesslyn Saw[1], Hannah McCalmont[3], Veronique Litalien[1], Jacqueline Boyle[1], Stefan E. Sonderegger[1], Ngoc Chau[4], Kathryn Evans[3], Loretta Cerruti[1], Jessica M. Salmon[1], Adam McCluskey [ID] [5], Richard B. Lock[3], Phillip J. Robinson [ID] [4], Stephen M. Jane[1,2] & David J. Curtis [ID] [1,2]

Intensive chemotherapy for acute leukemia can usually induce complete remission, but fails in many patients to eradicate the leukemia stem cells responsible for relapse. There is accumulating evidence that these relapse-inducing cells are maintained and protected by signals provided by the microenvironment. Thus, inhibition of niche signals is a proposed strategy to target leukemia stem cells but this requires knowledge of the critical signals and may be subject to compensatory mechanisms. Signals from the niche require receptor-mediated endocytosis, a generic process dependent on the Dynamin family of large GTPases. Here, we show that Dynole 34-2, a potent inhibitor of Dynamin GTPase activity, can block transduction of key signalling pathways and overcome chemoresistance of leukemia stem cells. Our results provide a significant conceptual advance in therapeutic strategies for acute leukemia that may be applicable to other malignancies in which signals from the niche are involved in disease progression and chemoresistance.

[1] Australian Centre for Blood Diseases, Central Clinical School, Monash University, Melbourne, VIC, Australia. [2] Department of Clinical Haematology, Alfred Health, Melbourne, VIC, Australia. [3] Lowy Cancer Research Centre, Children's Cancer Institute, University of New South Wales, Sydney, NSW, Australia. [4] Cell Signalling Unit, Children's Medical Research Institute, Sydney, NSW, Australia. [5] Chemistry, Centre for Chemical Biology, School of Environmental and Life Sciences, University of Newcastle, Callaghan, NSW, Australia. ✉email: cedric.tremblay@monash.edu

The hierarchical model posits that acute leukemias arise from leukemia stem cells (LSCs), which display stem cell-like properties that include long-term self-renewal and differentiation to generate the heterogeneity observed in the tumour at diagnosis[1]. Elimination of LSCs and their ancestral clones, pre-leukemic stem cells (pre-LSCs), is critical for long-term cure, as they are the source of relapse following chemotherapy[2–4]. How these cells survive high-dose chemotherapy remains poorly defined, but may include quiescence and pro-survival signals provided by the microenvironment[5–8]. To study mechanisms of therapeutic resistance in pre-LSCs, we have utilized the *Cd2-Lmo2*-transgenic (*Lmo2*[Tg]) mouse model of T-cell acute lymphoblastic leukemia (T-ALL)[9]. Using this model, we can identify and purify pre-LSCs with long-term self-renewal and resistance to high-dose therapeutic regimens[5,10]. Given that pre-LSCs harbour some, but not all the genetic lesions found at diagnosis, these cells are more reliant on the signals from the microenvironment to develop, self-renew and clonally evolve[11]. On the other hand, LSCs harbour the whole repertoire of genetic lesions found in overt leukemia, confirming the importance of activating mutations of growth factor signalling pathways in disease progression[12–14]. Although several studies have investigated the importance of the microenvironment during disease progression and chemoresistance[15–19], the catalogue of signals that mediate survival of relapse-inducing cells remains incomplete.

Several small molecules have been developed for targeting key signalling pathways that underpin the stem cell-like properties of LSCs as a therapeutic strategy in leukemia. Although potent inhibitors of specific signalling pathways, such as interleukin-7 (IL-7), Notch1 and Kit, displayed promising anti-leukemic activity in murine models[20–22], clinical studies revealed that long-term exposure to these small molecules was associated with substantial toxicity and therapeutic resistance[23,24]. The efficacy of these specific inhibitors may be limited by the inherent plasticity of LSCs, which selects for activation of alternative signalling pathways[25–27]. Therefore, targeting common components of multiple signalling pathways may counteract LSC plasticity and limit therapeutic resistance.

Dynamins are a family of large GTPases required for the budding and scission of the endosome during clathrin-mediated endocytosis, a fundamental process that regulates cell migration, cytokinesis, signal transduction, transport of nutrients and the recycling or degradation of proteins[28–30]. Although Dynamin 2 is ubiquitously expressed, Dynamin 1 is neuron-specific and Dynamin 3 is prominently found in the brain and testes. Using dominant-negative forms of Dynamin or small molecule inhibitors, studies showed that Dynamin-dependent endocytosis (DDE) is essential for signal transduction downstream of ligand-bound receptors activated by cues from the microenvironment[28]. Despite the development of several small molecule inhibitors to study DDE[31–34], their use as anti-cancerous drugs remains very limited, with a single study showing that inhibition of DDE enhanced target availability of antibody-dependent cellular toxicity in cancer cells[35]. Given the essential role of DDE in the regulation of cellular responses to stimuli produced by the microenvironment, we postulated that Dynamin inhibitors could impair multiple signalling pathways required for the growth and survival of LSCs. To this end, we tested 2-Cyano-*N*-octyl-3-(1-(3-dimethylaminopropyl)-1*H*-indol-3-yl)-acrylamide (Dynole 34-2), a specific and potent inhibitor of Dynamin GTPase activity in relevant models of acute leukemia.

## Results

### Small molecule inhibitors of Dynamin induce death of growth factor-dependent cells.
The IL-7 and Notch1 signalling pathways are prime therapeutic targets for T-ALL, although current agents are limited by acquired resistance or toxicity[12,26,36]. Given the important role of Dynamin for signal transduction, we investigated the effects of small molecule inhibitors of Dynamin on a factor-dependent Ba/F3 cell line expressing the receptor for IL-7 (Ba/F3-IL7R)[12]. The Dynamin inhibitors Dynasore, 3-Hydroxy-*N*'-[(2,4,5-trihydroxyphenyl)-methylidene]-naphthalene-2-carbohydrazide (Dyngo 4a) and Dynole 34-2 showed a dose-dependent inhibition of IL-7 signalling as measured by loss of phospho-Stat5 (pStat5; Fig. 1a and Supplementary Fig. 1a). Inhibition of pStat5 correlated with a dose-dependent accumulation of IL-7R at the surface of treated cells (Fig. 1b and Supplementary Fig. 1b). In accordance with their reported inhibitory activity[31,32], Dynole 34-2 was the most potent inhibitor of IL-7 signalling (IC$_{50}$: 4.29 ± 0.77 μM; Fig. 1a and Supplementary Fig. 1a) and therefore was used for all subsequent studies. Consistent with the IL-7 dependence of Ba/F3-IL7R cells, Dynole 34-2 induced cell death in a dose-dependent manner that correlated with inhibition of pStat5 (Supplementary Fig. 1c).

Previous studies have demonstrated that DDE is essential for IL-7 signal transduction[14,37,38], suggesting that Dynole 34-2 would prevent IL-7R internalization by impairing the endocytic process. To assess this hypothesis, we analysed the dynamic cellular localization of IL-7R, clathrin and the early endosomal marker Rab5 in Ba/F3-IL7R cells following IL-7 stimulation. In accordance with previous work on the dynamics of IL-7R internalization in normal and malignant T cells[14,37,38], we observed increased colocalization with clathrin (Fig. 1d and Supplementary Fig. 1d) and with Rab5-positive early endosomes within 5 min of IL-7 stimulation (Fig. 1e and Supplementary Fig. 1d). We observed no significant difference in the colocalization of IL-7R and clathrin in the presence of Dynole 34-2 (Fig. 1d and Supplementary Fig. 1d). In contrast, colocalization of IL-7R with Rab5 was markedly reduced in Ba/F3-IL7R cells treated with Dynole 34-2 (Fig. 1e and Supplementary Fig. 1d), suggesting that Dynole 34-2 impairs the internalization and subsequent distribution of IL-7R in early endosomes upon IL-7 stimulation. Consistent with the importance of IL-7R internalization for IL-7-mediated survival, cell death induced by Dynole 34-2 correlated with a dose-dependent accumulation of IL-7R at the surface of leukemic cells (Supplementary Fig. 1c).

In addition to blocking DDE, Dynamin inhibitors also inhibit cytokinesis, leading to apoptosis of growth factor-independent dividing cells[39,40]. To distinguish cell death caused by blocking DDE from effects on cytokinesis, we generated a Ba/F3-IL7R cell line expressing a doxycycline-inducible constitutively active form of Stat5 (Stat5-CA; Fig. 1f)[41]. Overexpression of Stat5-CA in BaF3 cells confers growth factor independence[41] and thus, Dynole 34-2 would not induce apoptosis in Ba/F3-IL7R[Stat5-CA] cells if the mechanism of action was through blocking signal transduction. In contrast, growth factor-independent Ba/F3-IL7R[Stat5-CA] cells should remain sensitive to Dynole 34-2 if cell death occurs primarily through inhibition of cytokinesis. We first confirmed that doxycycline treatment of Ba/F3-IL7R[Stat5-CA] cells induced pStat5 (Supplementary Fig. 1e) and enabled growth in the absence of IL-7 (Supplementary Fig. 1f). Consistent with blocking DDE as the mechanism of inducing apoptosis, Ba/F3-IL7R[Stat5-CA] were resistant to Dynole 34-2 in the presence of doxycycline (Fig. 1g). Similarly, cells with constitutive pStat5 were resistant to the treatment of the JAK1/2 inhibitor Ruxolitinib, which also acts upstream of Stat5. In contrast, Vincristine, a potent inhibitor of cytokinesis in cancer cells[42], was equally effective in killing Ba/F3-IL7R[Stat5-CA] cells in the presence or absence of Stat5-CA (Fig. 1g). Thus, Dynole 34-2 triggers apoptosis of factor-dependent cells by blocking DDE rather than cytokinesis.

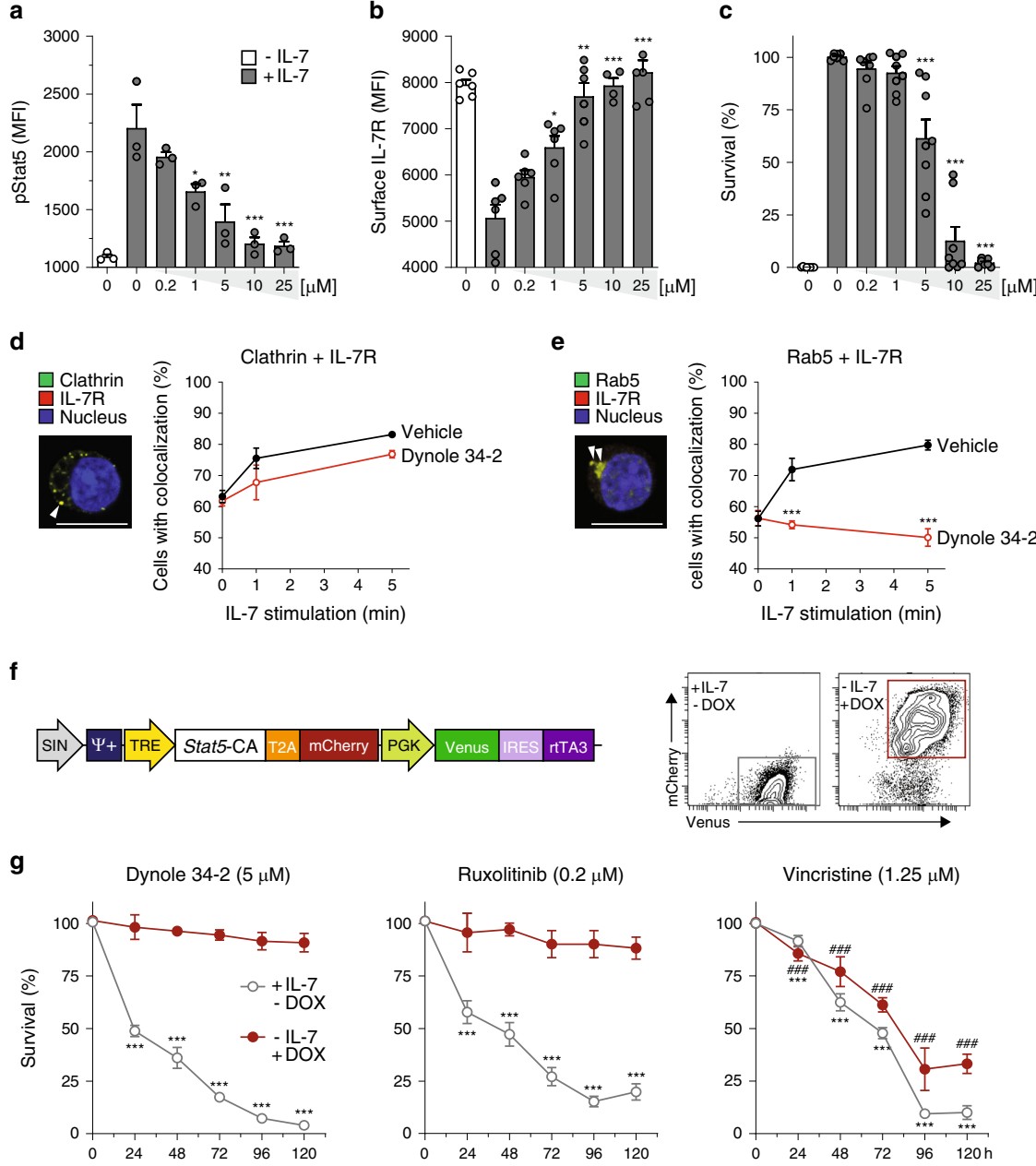

**Fig. 1 Dynole 34-2 induces apoptosis by blocking Dynamin-dependent endocytosis in IL-7-dependent Ba/F3 cells. a–c** Levels of phospho-Stat5 (pStat5; **a**), surface IL-7R (**b**) and survival (**c**) of Ba/F3-IL7R cells with increasing doses of Dynole 34-2, assessed by flow cytometry. Mean ± SD from three independent experiments are shown (*$P < 0.05$, **$P < 0.01$, ***$P < 0.001$ compared to vehicle + IL-7). Viability normalized to vehicle + IL-7 control (grey). Baseline is untreated cells without IL-7 (white bar). **d**, **e** Representative immunofluorescence staining for IL-7R colocalization with clathrin (**d**) and the early-endosome marker Rab5 (**e**) in Ba/F3-IL7R cells stimulated with IL-7. Nuclei were stained using DAPI and colocalization clusters were indicated with a white arrow. Proportion of cells presenting colocalization of IL-7R with either clathrin (**d**) or Rab5 (**e**), at different time points following IL-7 stimulation, were determined by reporting the number of cells displaying at least one colocalized staining for green and red fluorescence to the total number of cells, as previously described[14,38]. Scale bars: 10 μm. Mean ± SEM from three independent experiments (Supplementary Fig. 1d). Two-way ANOVA test with Sidak's correction; ***$p < 0.001$, as compared with Vehicle. **f** Schematic representation of the TxTCPVIR retroviral vector used for the doxycycline-inducible expression of constitutively active Stat5 (*Stat5*-CA) in Ba/F3-IL7R cells. The inducible expression of the transgene in Ba/F3-IL7R[Stat5-CA] cells can be assessed by measuring levels of mCherry by flow cytometry (right panels). **g** Cytotoxic activity of Dynole 34-2, Ruxolitinib or Vincristine in IL-7-dependent (+IL-7 −DOX; grey) and IL-7-independent (−IL-7 +DOX; red) Ba/F3-IL7R[Stat5-CA] cells. The doses used correspond to the median inhibitory (IC$_{50}$) or lethal concentration (LC$_{50}$) for each drug, determined in Ba/F3-IL7R cells cultured with IL-7 (Fig. 1a–c and Supplementary Fig. 1g, h). Viability is normalized to 100% at the time of initial drug treatment (day 0). Mean ± SD from three independent experiments are shown (***$P < 0.001$ compared to vehicle +IL-7 −DOX; ###$P < 0.001$ compared to vehicle −IL-7 +DOX.

**Dynole 34-2 blocks endocytosis and growth factor-induced signalling in pre-LSCs**. Environmental cues produced by the niche like Notch1 and IL-7 signalling play a crucial role in T-cell leukemogenesis[13,14,43]. Therefore, we postulated that Dynole 34-2 might perturb the growth of pre-LSCs, which rely upon growth factors from their microenvironment. To examine this hypothesis, we utilized the $Lmo2^{Tg}$ mouse model of T-ALL, in which the pre-LSCs activity is restricted to the CD4$^-$ CD8$^-$ CD44$^-$ CD25$^+$ CD28$^{low}$ (DN3a) population of T-cell progenitors[10,14]. In vitro growth of pre-leukemic thymocytes requires Kit-ligand (stem cell factor, SCF), Flt3 and IL-7, as well as a stromal cell line over-expressing the Notch1 ligand Delta-like 1 (DL1)[5,13,44]. Treatment of cultured $Lmo2^{Tg}$ DN3a cells with Dynole 34-2 significantly decreased the levels of pStat5 and Hes1—the canonical effector of Notch1 signalling (Fig. 2a). We also observed decreased phosphorylation of Erk, S6 and Akt, indicating that Dynole 34-2 can inhibit multiple signalling pathways in pre-LSCs. To confirm that Dynole 34-2 impaired IL-7 and Notch1 signalling pathways in pre-LSCs by preventing DDE, we examined receptor internalization following stimulation with IL-7 and DL1, respectively. Similar to Ba/F3-IL7R cells, Dynole 34-2 completely blocked the internalization of IL-7R and downstream activation of Stat5 in DN3a thymocytes (Fig. 2b). Dynole 34-2 also impaired Notch1 internalization and the downstream activation of Hes1 in DN3a cells stimulated with DL1 (Fig. 2c).

Ruxolitinib and the γ-secretase inhibitor N-[N-(3,5-Difluorophenacetyl)-L-alanyl]-S-phenylglycine t-butyl ester (DAPT), a small molecule that potently blocks Notch1 signalling, have activity in T-ALL[13,21]. Therefore, we compared the effects of Dynole 34-2 with Ruxolitinib and DAPT on pre-LSCs. Consistent with the factor-dependency of pre-LSCs, all three small molecules potently killed co-cultured $Lmo2^{Tg}$ DN3a cells in a dose-dependent manner (Fig. 2d and Supplementary Fig. 2a). Of note, wild-type (WT) DN3a thymocytes were fivefold less sensitive to Dynole 34-2 than pre-LSCs (Supplementary Fig. 2a, b). Importantly, Dynole 34-2 inhibited both pStat5 and Hes1 expression as effectively as Ruxolitinib for IL-7 and DAPT for Notch1 signalling (Supplementary Fig. 2c). Bone marrow suppression is a major limitation for the clinical use of Ruxolitinib[23,24]. Consistent with this effect, clonogenic assays using bone marrow cells showed that Ruxolitinib but not Dynole 34-2 significantly impaired the growth of normal hematopoietic stem and progenitor cells (HSPCs) at concentrations comparable to those needed to impair pre-LSCs (Supplementary Fig. 2d). These results suggest there is a wider therapeutic window for Dynole 34-2 compared with Ruxolitinib. Pre-LSC activity is assessed by their ability to repopulate the thymus of sublethally irradiated animals[10]. Therefore, we transplanted equivalent numbers of $Lmo2^{Tg}$ DN3a thymocytes following in vitro treatment with Dynole 34-2, Ruxolitinib or DAPT for 3 days. Analysis 6 weeks post-transplant revealed that all three agents significantly impaired the ability of pre-LSCs to engraft and expand in the thymus of sublethally irradiated recipients (Fig. 2e). Taken together, our results suggest that blocking DDE of multiple receptors with Dynole 34-2 significantly impairs pre-LSC activity as effectively as Ruxolitinib and DAPT.

Although we have demonstrated sensitivity of pre-LSCs to Dynole 34-2, the resistance of Ba/F3-IL7R$^{Stat5-CA}$ cells (Fig. 1e) suggests that LSCs carrying activating mutations of IL-7 or Notch1 signalling pathways may also be resistant. Therefore, we crossed our $Lmo2^{Tg}$ mouse model with transgenic mice expressing either a constitutively activated form of STAT5 ($STAT5-CA^{Tg}$)[41] or a hyperactive allele of Notch1 ($N1-ICD^{Tg}$)[45] to directly examine the sensitivity of DN3a thymocytes in the setting of constitutively activated signalling. As expected, $Lmo2^{Tg};STAT5-CA^{Tg}$ DN3a thymocytes were more resistant to

Ruxolitinib than $Lmo2^{Tg}$ DN3a cells by 5.3-fold and $Lmo2^{Tg};N1-ICD^{Tg}$ DN3a cells were 9-fold more resistant to DAPT than $Lmo2^{Tg}$ thymocytes (Supplementary Fig. 2a, b). Treatment of these cells with Dynole 34-2 showed that mutations downstream of DDE also conferred relative resistance (3.4-fold, Supplementary Fig. 2a, b), with survival similar that seen in WT DN3a thymocytes (Supplementary Fig. 2e). This decreased sensitivity of WT DN3a cells to Dynole 34-2 could be explained by the enhanced survival associated with increased levels of pStat5, as compared to $Lmo2^{Tg}$ thymocytes[14,46]. Of note, the presence of $STAT5-CA^{Tg}$ also conferred resistance to DAPT, and $Lmo2^{Tg};N1-ICD^{Tg}$ DN3a thymocytes were more resistant to Ruxolitinib, confirming that activation of alternative signalling pathways can promote resistance to different targeted therapies. Thus, Dynole 34-2 may be less effective for relapse-inducing cells that have acquired activating mutations downstream of DDE.

**In vivo treatment with Dynole 34-2 impairs pre-LSC activity.** To assess the therapeutic potential of Dynole 34-2, we first measured serum levels following a single intraperitoneal dose (30 mg/Kg). Pharmacokinetic analysis showed that Dynole 34-2 achieves an average maximum concentration ($C_{max}$) of >4 μM within 15 min, with an estimated half-life of 30 min (Supplementary Fig. 3a). To assess the in vivo activity of Dynole 34-2 on pre-LSCs, 6-week-old $Lmo2^{Tg}$ mice were administered 30 mg/kg intraperitoneally twice daily for 5 days on 2 consecutive weeks (Fig. 3a). Analysis of mice at the end of treatment revealed a tenfold reduction in the numbers of DN3a thymocytes (Fig. 3a), associated with significantly reduced pStat5 and Hes1 levels (Fig. 3b). Most importantly, limiting dilution transplantation of thymocytes from these treated mice revealed a 100-fold decrease in pre-LSC frequency (Fig. 3c). Serial transplantation is the gold-standard assay to assess self-renewal capacity[5]. Using this functional assay, we observed progressive exhaustion of pre-LSCs in mice treated with Dynole 34-2, with complete loss of repopulating activity in tertiary recipients (Supplementary Fig. 3b).

We have previously reported that long-term self-renewal and therapeutic resistance is limited to a rare population of cell cycle-restricted pre-LSCs[5]. Using our $TetOP-H2B-GFP^{KI/+};Lmo2^{Tg}$ ($H2B-GFP;Lmo2^{Tg}$) mouse model, we showed that a small fraction of DN3a thymocytes (1.7 ± 0.2%, Supplementary Fig. 3c) retained green fluorescent protein (GFP) expression (GFP$^{hi}$) 2 weeks after withdrawal of doxycycline following a 6-week labelling period, as previously established[5]. In this model, administration of Dynole 34-2 during the 2-week chase period did not affect the proportion of GFP$^{hi}$ cells in treated mice, suggesting that cell cycle-restricted DN3a cells did not display increased resistance to Dynole 34-2 (Supplementary Fig. 3c). Moreover, we observed a tenfold decrease in the number of DN3a GFP$^{hi}$ thymocytes in mice administered with Dynole 34-2 (Supplementary Fig. 3d). In accordance with published gene expression data from GFP$^{hi}$ and GFP$^{lo}$ DN3a thymocytes[5], levels of pStat5 and Hes1 were not affected by cell cycle kinetics (Supplementary Fig. 3e), although significantly decreased in DN3a cells from treated mice. Importantly, Dynole 34-2 significantly impaired repopulation activity and leukemogenicity of GFP$^{hi}$ DN3a thymocytes (Supplementary Fig. 3f–h). Altogether, this data confirms that Dynole 34-2 has significant in vivo activity on long-term self-renewing pre-LSCs as a single agent.

Activation of pro-survival signalling pathways is an adaptive mechanism of chemotherapy resistance[47]. To assess whether simultaneously blocking IL-7 and Notch1 signalling could overcome chemoresistance of pre-LSCs, we treated $Lmo2^{Tg}$ mice with the combination of Dynole 34-2 and chemotherapeutic agents (vincristine, dexamethasone and L-asparaginase, VXL)

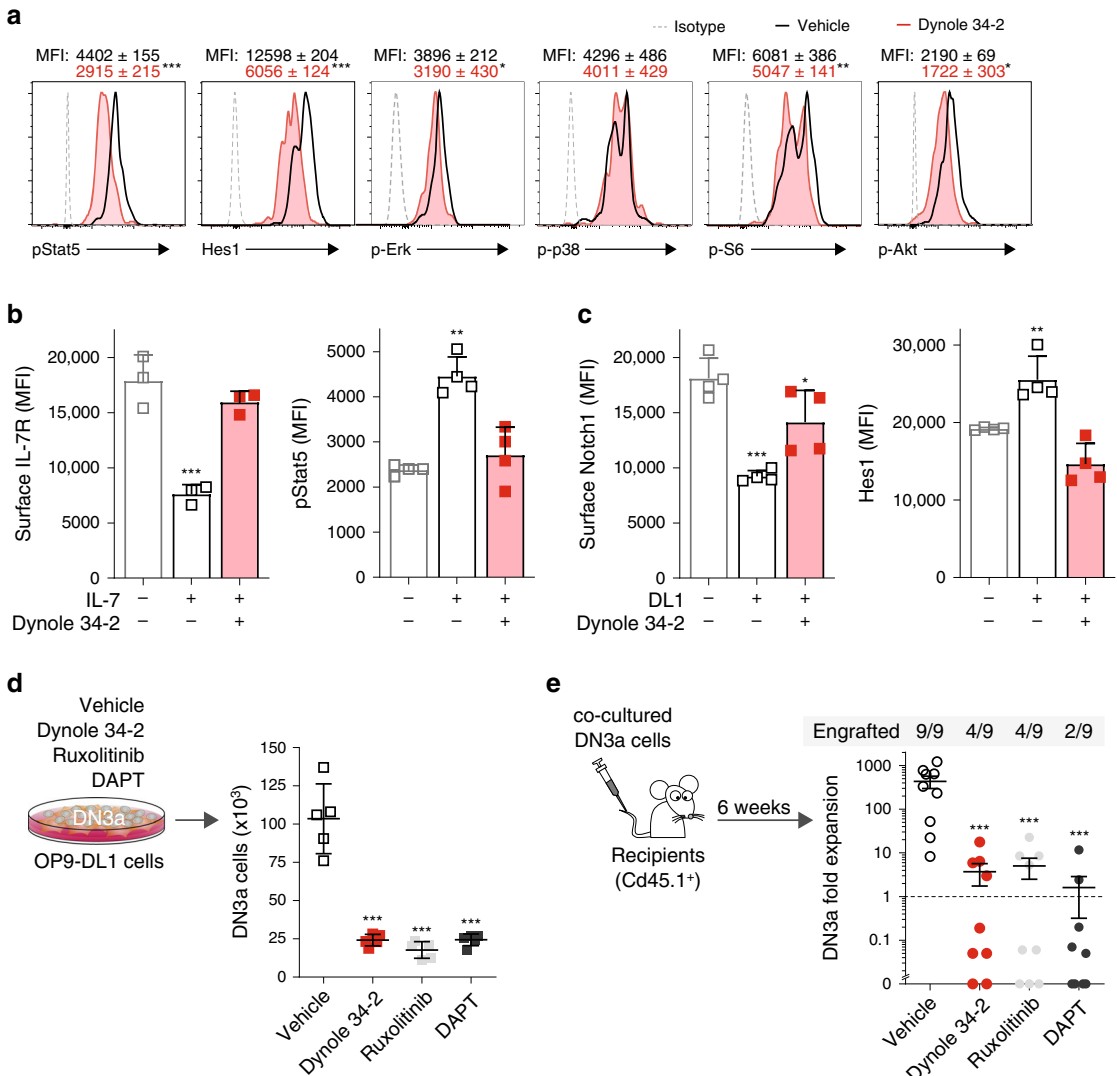

**Fig. 2 Dynole 34-2 prevents the activation of the signalling network downstream of IL-7R and Notch1 in pre-leukemic DN3a thymocytes. a** Levels of growth factor-mediated signalling effectors in *Lmo2*[Tg] DN3a cells treated with vehicle or Dynole 34-2 co-cultured on OP9-DL1 with cytokines, assessed by flow cytometry. Mean fluorescence intensity (MFI) ± SD of n = 4 individual animals analysed in duplicate are shown (Mann–Whitney test; *P < 0.05 and ***P < 0.001 compared to vehicle). b, Internalization assay for IL-7R (left) and downstream activation of Stat5 (pStat5; right) in *Lmo2*[Tg] DN3a cells in steady state and after stimulation with IL-7, in presence of 0.2 μM Dynole 34-2. MFI ± SD of n = 3 individual mice, performed in duplicate. Student's t-test, **P < 0.01, ***P < 0.001 compared to unstimulated cells. **c** Surface levels of Notch1 (left) and intra**c**ellular expression of Hes1 (right) in *Lmo2*-transgenic DN3a thymocytes, co-cultured overnight on OP9 or OP9-DL1 (DL1) cells in the presence of 0.2 μM Dynole 34-2. MFI ± SD of n = 3 individual animals, performed in duplicate. Student's t-test *P < 0.05, **P < 0.01, ***P < 0.001 compared to OP9 (unstimulated) controls. **d, e** Absolute numbers of co-cultured *Lmo2*[Tg] DN3a thymocytes treated in vitro with either vehicle, Dynole 34-2 (0.2 μM), Ruxolitinib (0.2 μM) and DAPT (0.1 μM) for 72 h (**d**), and in vivo fold expansion of these cells measured 6 weeks post-transplantation in the thymus of recipients (**e**). Mean ± SD, Student's t-test (N = 4 individual mice). ***P < 0.001 compared to v**e**hicle. Number of recipients engrafted is indicated.

typically used for T-ALL[48]. Analysis of total DN3a thymocytes numbers immediately after 2 weeks treatment showed a 17-fold reduction with combination therapy, which was significantly greater than either Dynole 34-2 or chemotherapy alone (Fig. 4a and Supplementary Fig. 4a). Of note, Dynole 34-2 showed preferential killing of DN3a cells compared to chemotherapy, which had no selectivity for the pre-leukemic T-cell progenitor population (Fig. 4b and Supplementary Fig. 4b). Given that self-renewal enables pre-LSCs to accumulate additional genetic events promoting clonal selection and progression to leukemia, we postulated that Dynole 34-2 and chemotherapy would affect clonality of pre-LSCs and their capacity of generating T-ALL. Consistent with this idea, there was reduced monoclonality in DN3a cells from *Lmo2*[Tg] treated with Dynole 34-2 or

chemotherapy (Supplementary Fig. 4c), as assessed by *Tcrβ* rearrangement[49]. Although chemotherapy did not affect the leukemogenic potential of pre-LSCs, we observed delayed progression to leukemia in recipients injected with thymocytes from *Lmo2*[Tg] treated with Dynole 34-2 (Supplementary Fig. 4d). Strikingly, the combination therapy eradicated pre-leukemic clones in DN3a cells (Supplementary Fig. 4c), which resulted in loss of leukemogenicity as five out of six of recipients transplanted with treated pre-LSCs remained leukemia-free for up to 7 months (Supplementary Fig. 4d). This drastic effect could be explained by the synergy between Dynole 34-2 and chemotherapy on pre-LSC self-renewal activity, where the combination generated 50-fold fewer donor-derived DN3a cells than chemotherapy alone (Fig. 4c).

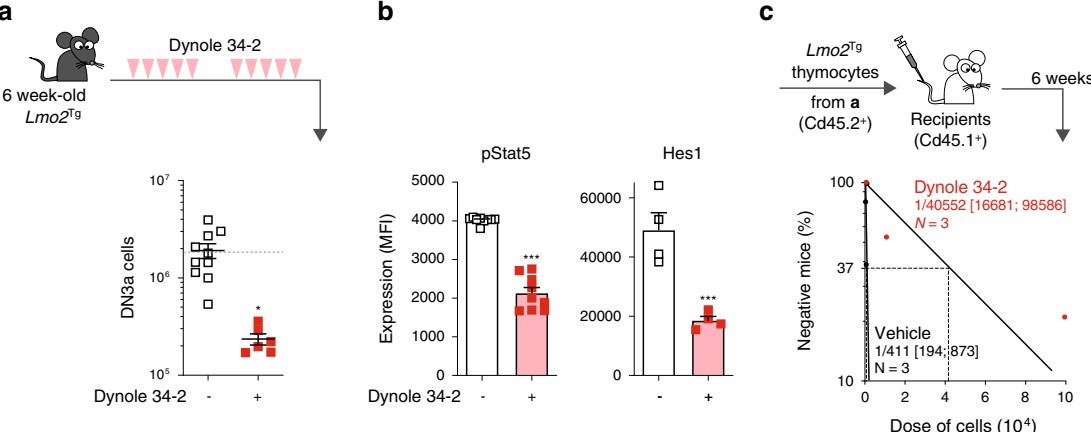

**Fig. 3 Inhibition of receptor-mediated endocytosis of IL-7R and Notch1 by Dynole 34-2 impairs pre-LSC self-renewal. a** Treatment schematic and absolute numbers of DN3a T-cell progenitors in the thymus of 6-week-old *Lmo2*[Tg] mice following treatment with vehicle or Dynole 34-2. Mean ± SEM, Student's *t*-test; *P < 0.05 compared to vehicle. Each square represents an individual animal. Dashed line indicated the mean absolute number of DN3a thymocytes in wild-type (WT) mice at 6 weeks of age. **b** Levels of activated Stat5 (pStat5, left) and Hes1 (right) proteins in DN3a cells from *Lmo2*[Tg] mice treated with vehicle or Dynole 34-2, assessed by flow cytometry. Mean ± SD of *n* = 3 individual mice are shown. Student's *t*-test ***P < 0.001 compared to vehicle. **c** Pre-LSC frequency within the DN3a thymocyte population of *Lmo2*[Tg] mice treated with vehicle or Dynole 34-2 assessed by limiting dilution assays. Mice were scored positive when T-cell lineage reconstitution was more than 1%, as previously described[13]. Pre-LSC frequencies and [95% confidence intervals] are shown and calculated from three individual mice.

The impaired engraftment of *Lmo2*[Tg] pre-LSCs treated with Dynole 34-2 might be explained by defects in cell homing, in which DDE has been implicated[29]. To exclude this possibility, we used a cell-marking strategy that enables detection of pre-LSCs in the absence of transplantation. In this assay, treatment of *Mx1-Cre* transgenic *Rosa26-YFP* (*Mx;YFP*) reporter mice with polyinosinic-polycytidylic acid (poly (I:C)) induces yellow fluorescent protein (YFP) expression in hematopoietic stem cells (HSCs)[10] (Supplementary Fig. 4e). In WT *Mx;YFP* mice, normal turnover of the thymus by HSCs leads to the generation of YFP-labelled DN3a cells over a 3-week period. However, in *Mx;YFP;Lmo2*[Tg] mice, the presence of self-renewing pre-LSCs prevents YFP-labelled HSCs to replenish the thymus[10] and, thus, *Lmo2*[Tg] DN3a thymocytes remain YFP negative. Therefore, we reasoned that therapies targeting pre-LSC self-renewal would allow migration of YFP[+] HSCs with the generation of YFP-labelled thymocytes. One week after poly (I:C) treatment, *Mx;YFP;Lmo2*[Tg] mice were treated with Dynole 34-2, chemotherapy or the combination, and then YFP expression in DN3a thymocytes was examined 3 weeks later, to allow thymic immigration of YFP[+] HSCs (Fig. 4d). As previously reported[10], >90% of DN3a thymocytes from vehicle-treated *Mx;YFP;Lmo2*[Tg] mice remained YFP-negative, confirming the presence of self-renewing pre-LSCs that prevent intrathymic settlement of YFP[+] HSCs (Fig. 4e). Furthermore, chemotherapy alone had no significant effect, confirming the intrinsic chemoresistance of pre-LSCs. In contrast, Dynole 34-2 alone increased the number of YFP-labelled DN3a thymocytes from 10% to 46% (Fig. 4e). Most striking was the effect of Dynole 34-2 combined with chemotherapy, where the thymus of *Mx;YFP;Lmo2*[Tg] mice was almost completely replenished by YFP[+] cells (Fig. 4e). Importantly, the combination of Dynole 34-2 and chemotherapy was well tolerated with no detrimental effect on mouse weight (Supplementary Fig. 4f).

We then assessed the safety of the combination of Dynole 34-2 and chemotherapy in non-tumour-bearing mice, as the effects of these drugs on normal hematopoiesis have not been reported. As a single agent, Dynole 34-2 had no detrimental effect on differentiated cells in the thymus and the bone marrow of treated mice (Supplementary Fig. 5a–c), or the numbers of phenotypic

bone marrow stem and progenitor cells (Supplementary Fig. 5d). On the other hand, chemotherapy significantly decreased the absolute numbers of most progenitor and differentiated cell populations analysed (Supplementary Fig. 5b–d). To functionally assess the effect of these drugs on stem cell activity, we performed a competitive transplant with total bone marrow cells from treated mice (Supplementary Fig. 5e). Unlike chemotherapy, which significantly impairs HSC activity[50–52], Dynole 34-2 had no detrimental effect on HSC fitness and differentiation potential (Supplementary Fig. 5f–h). Importantly, Dynole 34-2 prevented the chemotherapy-induced decrease of HSC activity and restored normal differentiation in recipients, suggesting a protective effect of Dynole 34-2 in genotoxic stress conditions. Thus, Dynole 34-2 overcomes chemoresistance of pre-LSCs without detrimental effects on normal hematopoietic stem cells and progenitors.

**Dynole 34-2 sensitizes LSCs to chemotherapy and delays the onset of leukemia.** To study the effect of Dynole 34-2 on the LSC population residing within established T-ALL, we first defined the T-cell population enriched for LSC activity in the *Lmo2*-transgenic model. Three independent T-ALL samples arising from *Lmo2*[Tg] mice were fractionated into immature DN (DN1, DN3a and DN4) sub-populations, as well as ISP8 and DP populations for transplant into sublethally irradiated recipients (Fig. 5a). In all T-ALL samples tested, the LSC activity was substantially enriched in the DN3a population (Fig. 5b), confirming that LSCs do not phenotypically shift during the leukemogenic process. Therefore, we focused our analyses of the effects of Dynole 34-2 on the DN3a fraction in T-ALL samples. Similar to pre-LSCs, Dynole 34-2 blocked downstream signalling activation by IL-7 and Notch1 in DN3a cells derived from T-ALL samples (Fig. 5c). Although LSCs were more resistant than pre-leukemic *Lmo2*[Tg] DN3a thymocytes (Supplementary Figs. 2a, b, 6a), the ability of Dynole 34-2 to inhibit pStat5 and Hes1 led to a dose-dependent killing of LSCs despite the presence of activating mutations of either IL-7 or Notch1 signalling pathway (Supplementary Fig. 6a, b). In contrast, Ruxolitinib and DAPT were more selective for exclusively inhibiting IL-7 or Notch signalling, respectively (Supplementary Fig. 6c, d). Thus, Dynole 34-2 can

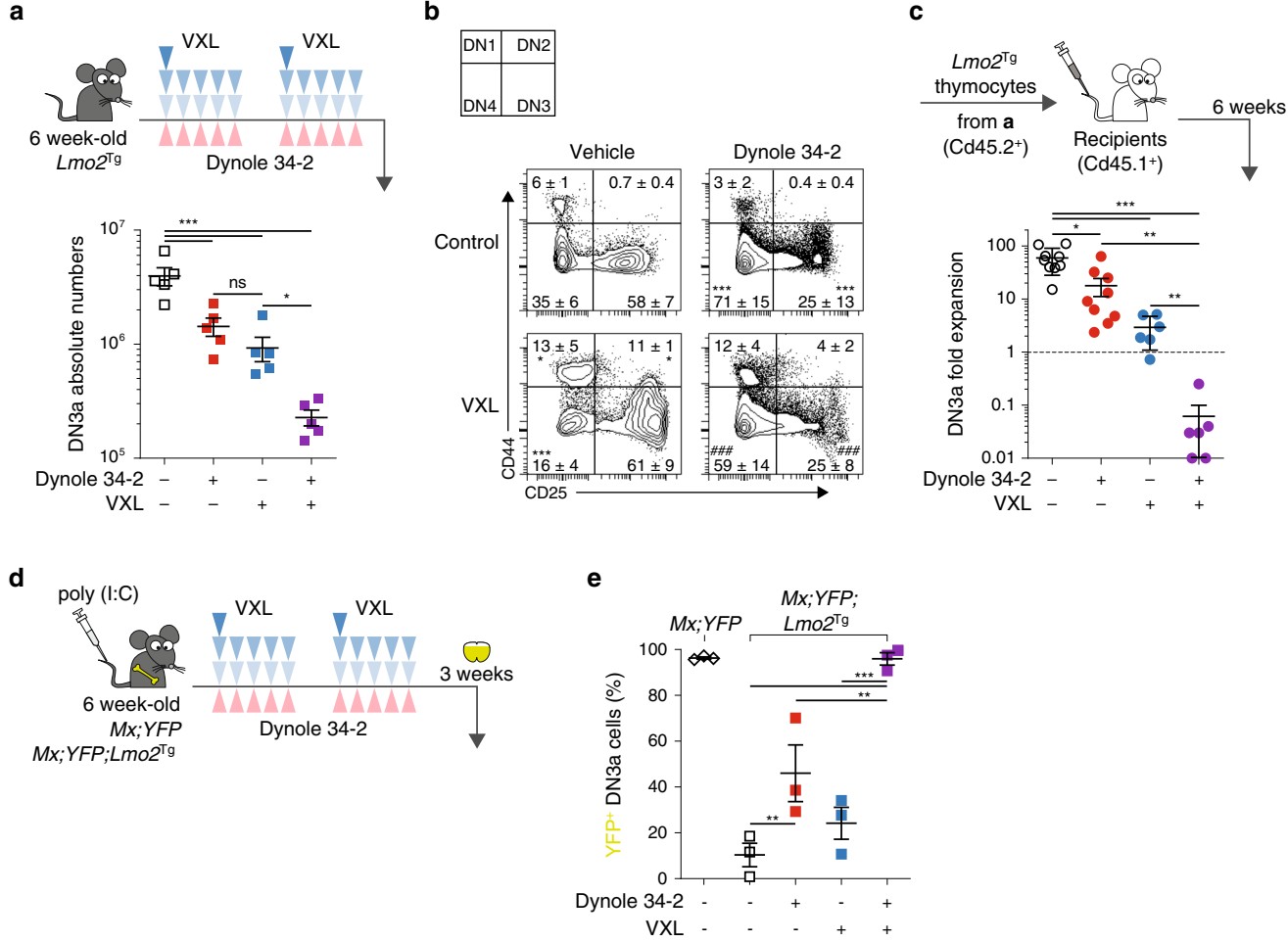

**Fig. 4 Dynole 34-2 sensitizes pre-LSCs to induction-like therapy. a** Treatment schematic and absolute numbers of DN3a T-cell progenitors in the thymus of 6-week-old *Lmo2*[Tg] mice following administration of two rounds of vehicle and Dynole 34-2, alone or combined with VXL chemotherapy. Mean ± SEM, two-way ANOVA with Tukey's correction test; *$P < 0.05$ and ***$P < 0.001$. Each square represents an individual animal. **b** Representative flow cytometry analysis of the CD4−CD8− (DN) T-cell progenitor populations in the thymus of 6-week-old *Lmo2*[Tg] mice following administration of two rounds of vehicle and Dynole 34-2, alone or combined with VXL chemotherapy. Mean ± SD ($N = 5$ individual mice), two-way ANOVA with Bonferroni correction test; *$P < 0.05$ and ***$P < 0.001$ compared to vehicle; ###$P < 0.001$, as compared to VXL, respectively. **c** Fold expansion of DN3a thymocytes from *Lmo2*[Tg] mice treated with either vehicle, Dynole 34-2, VXL or combination therapy with Dynole 34-2. Mean ± SEM, two-way ANOVA with Bonferroni correction test *$P < 0.05$, **$P < 0.01$ and ***$P < 0.001$. Each circle represents an individual animal. **d** Scheme for the intrathymic competition assays in the *Mx;YFP* and *Mx;YFP;Lmo2*[Tg] mice, which were injected with poly (I : C) to induce the expression of YFP in the bone marrow stem cells (HSCs), which can be tracked in the thymus 3 weeks post-injection. **e** Proportion of HSC-derived (%YFP+) DN3a thymocytes in *Mx;YFP;Lmo2*[Tg] mice treated with vehicle, Dynole 34-2, VXL and the combination Dynole 34-2 + VXL. *Mx;YFP* mice were used as positive controls. Mean ± SEM ($N = 3$ individual mice), two-way ANOVA with Tukey's correction test; *$P < 0.05$, **$P < 0.01$ and ***$P < 0.001$.

inhibit pathologically relevant signalling pathways in LSCs from T-ALL despite mutually exclusive activating mutations of IL-7 or Notch1.

To test the in vivo efficacy of Dynole 34-2 in T-ALL, mice injected with *Lmo2*[Tg] primary leukemias cells were treated with Dynole 34-2, chemotherapy or the combination (Fig. 5d). Analysis 24 h following the last dose demonstrated significant benefit for the combination therapy with 10- to 100-fold less *Lmo2*[Tg] DN3a leukemic cells in the thymus, bone marrow, spleen and peripheral blood compared with chemotherapy or Dynole 34-2 alone (Fig. 5e, f and Supplementary Fig. 6e). Consequently, Dynole 34-2 combined with chemotherapy significantly improved long-term survival of mice, with some recipients free of disease 16 weeks after treatment (Fig. 5g and Supplementary Fig. 6f, g). Altogether, these results demonstrate a therapeutic benefit for the combination of Dynole 34-2 and chemotherapy in *Lmo2*-induced T-ALL.

**Dynole 34-2 is efficacious in human T-ALL.** T-ALL is a highly heterogeneous disease subclassified according to the stage of thymic maturation, from the early T-cell precursor T-ALL (ETP-ALL) to the more mature subtypes[53]. Activating mutations of growth factor-induced signalling pathways, including IL-7 and NOTCH1, are present in two-thirds of T-ALL cases[53,54]. To determine whether Dynamin inhibitors exhibited subtype-specific efficacy in human T-ALL, we treated patient-derived ETP-ALL and mature T-ALL xenografts[21,55] with Dynole 34-2 for an hour prior to in vitro stimulation with growth factors. Similar to mouse T-ALL cells, Dynole 34-2 significantly decreased pSTAT5 and HES1 protein levels in both immature (Fig. 6a) and mature (Fig. 6b) subtypes of T-ALL. This inhibition of signalling led to a dose-dependent killing, irrespective of the presence of IL-7 or NOTCH1 pathway mutations (Supplementary Fig. 7a, b and Table 1).

We evaluated the preclinical potential of Dynole 34-2 with xenograft models of different subtypes of human T-ALL, using

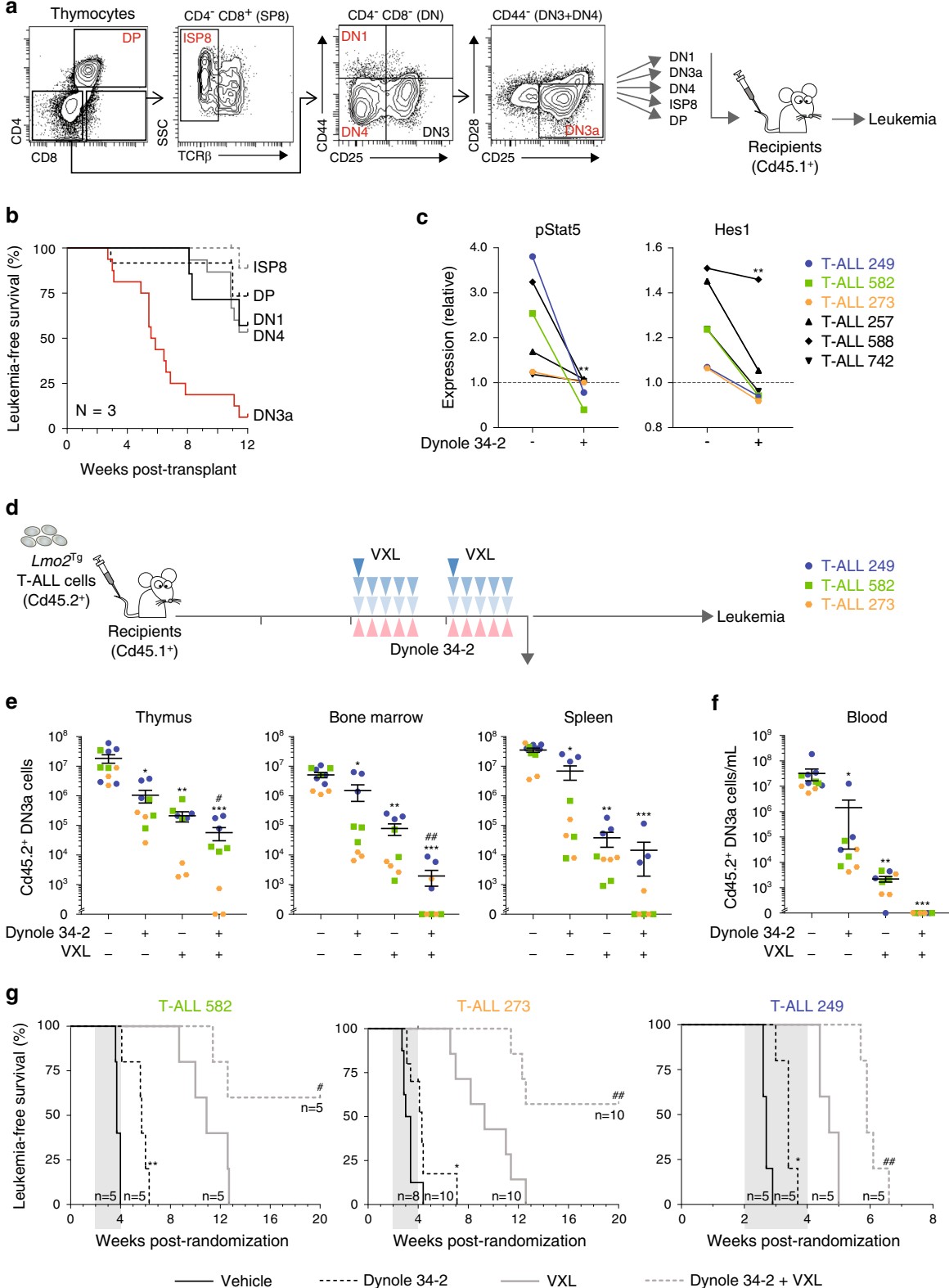

ETP12 and ALL8 cells, as these patient-derived samples demonstrated in vitro response to growth factors that could be inhibited by Dynole 34-2 (Fig. 6a, b). As represented in Fig. 6c, recipients were injected with patient-derived xenografts, randomized after engraftment was confirmed in the peripheral blood, and subsequently administered with either vehicle or Dynole 34-2, as a single agent or in combination with VXL chemotherapy. In

vivo, Dynole 34-2 showed single agent activity in both immature (ETP12) and mature (ALL8) T-ALL, resulting in significant reduction in leukemic cells in the peripheral blood, bone marrow and spleen (Fig. 6d, e and Supplementary Fig. 7c, d). Inhibition of IL-7 and NOTCH1 signalling pathways was confirmed in patient-derived cells 24 h after the last administration of Dynole 34-2 (Supplementary Fig. 7e, f). These positive effects on leukemic cells

**Fig. 5 Efficacy of Dynole 34-2 against LSCs from primary *Lmo2*-transgenic T-ALL. a** Flow sorting and schematic representation of the transplantation strategy of purified leukemic populations from three different primary *Lmo2*[Tg] thymomas. Indicated leukemic populations were purified and $5 \times 10^4$ cells were transplanted into sublethally irradiated Cd45.1[+] recipients. **b** Kaplan–Meier curves of mice injected with the purified populations (DN1, $n = 5$; DN3a, $n = 17$; DN4, $n = 15$; ISP8, $n = 10$; DP, $n = 12$) from three different *Lmo2*[Tg] primary leukemias. **c** Relative levels of activated Stat5 (pStat5) and Hes1 in leukemic DN3a cells from primary *Lmo2*[Tg] T-ALL treated with Dynole 34-2, after in vitro stimulation of either the IL-7 (left) or Notch1 (right) signalling pathway. Primary leukemias are indicated on the right with T-ALL 249 in blue, T-ALL 582 in green, T-ALL 573 in orange, and other leukemias are in black; unstimulated leukemic DN3a cells treated with vehicle were used as control, and reported as 1 (dashed line). Mean ± SEM ($N = 6$ primary leukemias, performed in duplicate), two-way ANOVA with Bonferroni correction test; **$P < 0.01$ as compared to vehicle. **d** Experimental setting for testing the efficacy of Dynole 34-2, VXL and combination therapy in sublethally irradiated Cd45.1[+] recipients injected with *Lmo2*[Tg] primary leukemias. **e, f** Relative numbers of leukemic DN3a cells in the thymus, bone marrow, spleen (**e**) and blood (**f**) of recipients, 24 h after the last administration of Dynole 34-2, VXL and combination therapy. Absolute numbers of DN3a cells in recipients treated with vehicle were used as control (dashed line) and reported as 100%. Mean ± SEM ($N = 3$ recipients per individual leukemia, analysed in duplicate), two-way ANOVA with Tukey's correction test; *$P < 0.05$, **$P < 0.01$ and ***$P < 0.001$ compared to vehicle; ##$P < 0.01$ and ###$P < 0.001$ compared to VXL. **g** Kaplan–Meier curves of sublethally irradiated recipients injected with *Lmo2*[Tg] primary T-ALL, treated with Dynole 34-2, VXL or combination therapy. Log-rank (Mantel–Cox) test; *$P < 0.05$ and **$P < 0.01$ compared to vehicle; #$P < 0.05$ and ##$P < 0.01$ compared to VXL. The period of administration is indicated in light grey, with the number of recipients for each cohort indicated for each primary leukemia.

burden and growth factor-induced signalling led to a significant survival advantage in recipients treated with Dynole 34-2 as a single agent, and more strikingly when it was combined with chemotherapy (Fig. 6f, g and Supplementary Fig. 7g–i). Analysis performed 24 h following the last administration confirmed that Dynole 34-2 enhanced the efficacy of chemotherapy with at least tenfold less patient-derived leukemic cells in bone marrow and spleen of recipient mice, compared with chemotherapy alone (Fig. 6d, e and Supplementary Fig. 7c). Altogether, our results suggest that inhibition of DDE with Dynole 34-2 represents an effective therapeutic strategy for human T-ALL.

**Activity of Dynole 34-2 in human AML.** Growth factors secreted by the niche have been shown to promote therapeutic resistance and disease progression in several haematological malignancies, including acute myeloid leukemia (AML)[56,57]. In AML, LSCs emerge from HSPCs, which rely upon niche signals to develop and self-renew[1,58]. Importantly, the expression of the human orthologues of poorly cross-reacting cytokines SCF, granulocyte/macrophage-stimulating factor (GM-CSF) and IL-3 (SGM3) significantly improved the repopulation capacity of patient-derived AML xenografts[59], suggesting that these signals are important for LSCs in AML. To study the effects of blocking DDE on the growth factor-induced signalling pathways most relevant to AML, we generated Ba/F3 cells expressing the receptors for SCF and GM-CSF (Ba/F3-SGM3R). Consistent with growth factor-dependent survival of Ba/F3-SGM3R cells, Dynole 34-2 induced cell death in a dose-dependent manner that correlated with inhibition of cytokine-induced signalling, as measured by decreased pStat5 following GM-CSF and IL-3 stimulation, as well as SCF-induced pErk (Supplementary Fig. 8a–c). Importantly, Dynole 34-2 impaired endocytosis of c-Kit, GM-CSFR and IL-3R (Supplementary Fig. 8d). Taken together, our data confirm that Dynole 34-2 triggers apoptosis of factor-dependent cells by blocking multiple signalling pathways that are crucial for AML.

Increased growth factor-induced signalling has been associated with poor prognosis for AML patients[7,60], suggesting that blocking these pathways may be of clinical relevance. To address this question, we treated patient-derived AML samples with Dynole 34-2 prior to stimulation with cytokines. In vitro, Dynole 34-2 blocked downstream signalling in human AML cells, as measured by the levels of pSTAT5 (Fig. 7a and Supplementary Fig. 9a), and prevented the internalization of IL-3R and GM-CSFR in patient-derived AML cells stimulated with IL-3 and GM-CSF, respectively (Supplementary Fig. 9b). We also found that Dynole 34-2 prevented the internalization of KIT and the downstream activation of ERK in AML cells that responded to SCF stimulation

(Fig. 7a and Supplementary Fig. 9b). Therefore, Dynole 34-2 effectively prevents the activation of multiple growth factor-stimulated signalling pathways by blocking DDE in human AML.

To evaluate the efficacy of Dynamin inhibitors in human AML, we performed clonogenic assays with primary leukemias in presence of relevant cytokines. Dynole 34-2 significantly decreased colony numbers in a dose-dependent manner (Supplementary Fig. 9c). Despite the wide range of responses observed with patient-derived AML samples, normal human CD34-positive HSPCs were significantly more resistant to Dynole 34-2 (LC$_{50}$: $23.03 \pm 5.72$ μM, Fig. 7b). We observed similar sensitivity for AML samples carrying FLT3-ITD or activating mutations of *JAK2* (Supplementary Fig. 9d and Table 2), suggesting that reported STAT5-activating mutations were not limiting the efficacy of Dynole 34-2. Altogether, our results suggest that potent Dynamin inhibitors like Dynole 34-2 can inhibit multiple growth factor-induced signalling pathways that promote survival in AML.

We assessed the in vivo efficacy of Dynole 34-2 for human AML using xenograft models carrying oncogenic mutations of either FLT3 or KRAS, as these genetic lesions are amongst the most common activating mutations of growth factor-induced signalling pathways found in AML at diagnosis (around 25% and 15% of cases, respectively)[61]. Therefore, we used AML01-307 and AML18, as these patient-derived samples demonstrated in vitro response to growth factors that could be inhibited by Dynole 34-2. Immunodeficient mice engrafted with primary human AML cells were subsequently treated with Dynole 34-2, the chemotherapeutic agents Daunorubicin and Cytarabine (DA$^{5+3}$)[59,62] or the combination (Fig. 7c). Inhibition of IL-3, GM-CSF and SCF signalling pathways was confirmed in patient-derived cells 24 h after the last administration of Dynole 34-2 (Supplementary Fig. 9e, f). Analysis at this time point demonstrated significant benefit for the combination therapy with 10 to 100-fold less patient-derived AML cells in bone marrow and spleen of recipient mice compared with DA$^{5+3}$ chemotherapy or Dynole 34-2 alone (Fig. 7d, e and Supplementary Fig. 9g, h). Consequently, the combination of Dynole 34-2 and DA$^{5+3}$ achieved significant inhibition of tumour burden and delayed the onset of the disease in recipients, exceeding the effect achieved with either single agents (Fig. 7f, g and Supplementary Fig. 9i–k). Taken together, our data suggest that inhibition of DDE with Dynamin inhibitors represents an effective therapeutic strategy for both T-ALL and AML.

## Discussion

Dynamins control receptor-mediated endocytosis, vesicle recycling and actin-cytoskeletal dynamics, which are crucial for the

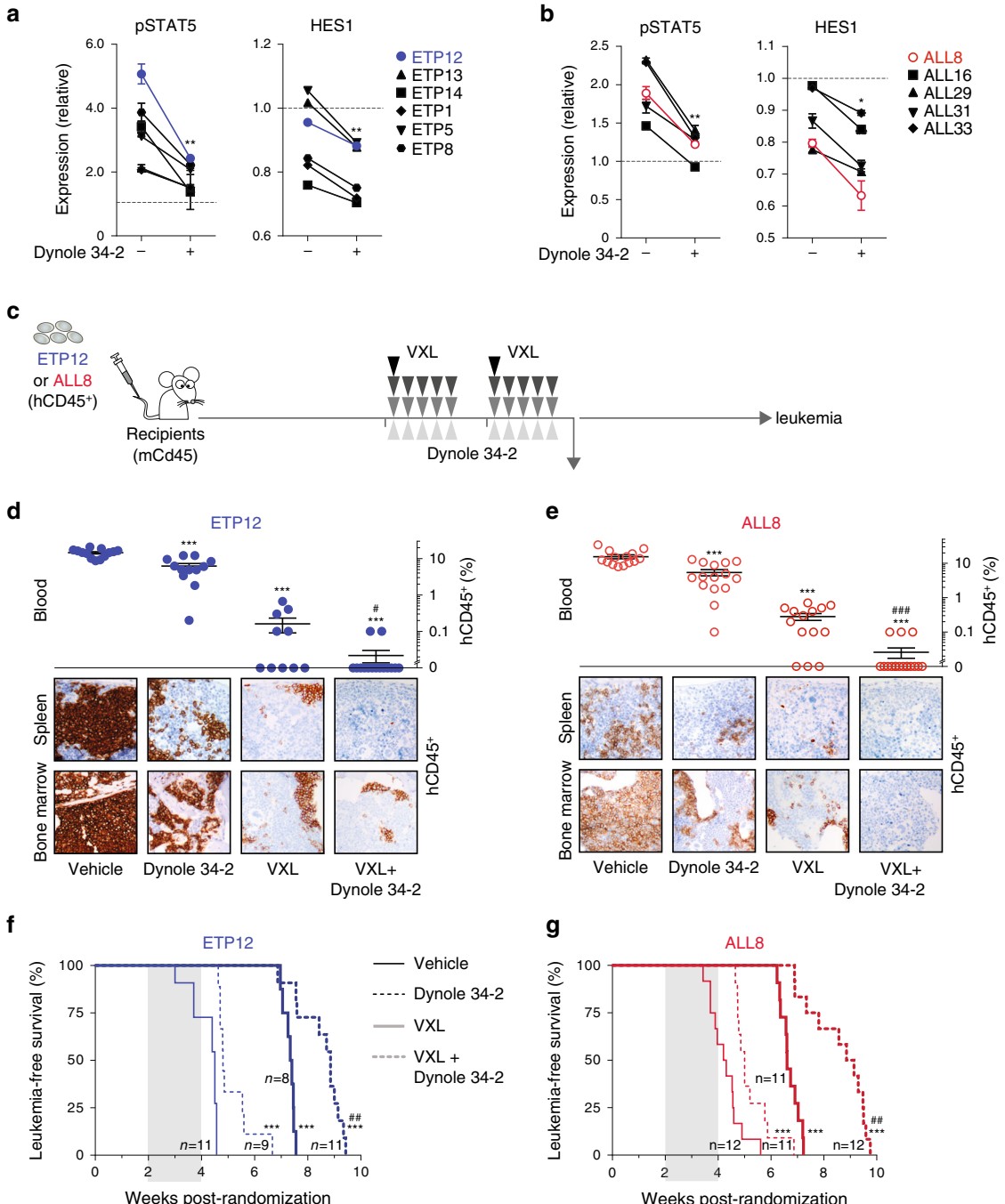

**Fig. 6 Dynole 34-2 is effective against human ETP-ALL and mature T-ALL. a, b** Relative levels of IL-7-induced activation of STAT5 (pStat5, left) and NOTCH1-induced expression of HES1 (right) in (**a**) ETP-ALL and (**b**) T-ALL cells from human xenografts treated with vehicle and Dynole 34-2, after in vitro stimulation. ETP12 is depicted with solid blue circles, ALL8 with red open circles and all other leukemias in black. Levels in unstimulated xenograft cells treated with vehicle were used as control, and reported as 1 (dashed line). Mean ± SD ($N = 6$ primary leukemias, performed in duplicate), two-way ANOVA with Bonferroni correction test; *$P < 0.05$ and **$P < 0.01$ as compared to vehicle. **c** Experimental setting for testing the efficacy of Dynole 34-2, VXL and combination therapy in xenograft models of human acute leukemia. Sublethally irradiated NSG recipients were randomized after engraftment was confirmed in the peripheral blood, and subsequently treated when the average proportion of human leukemic cells in the peripheral blood reached 1%. **d, e** Proportion of patient-derived cells (%hCD45[+]) in the peripheral blood (top), as well as immunochemistry against hCD45[+] leukemic cells in the bone marrow and spleen (bottom) of recipients injected with ETP12 (**d**) and ALL8 (**e**), 24 h after the last drug injection. Scale bars: 10 μm. Mean ± SEM, two-way ANOVA with Tukey's correction test; ***$P < 0.0001$ compared to vehicle; #$P < 0.05$ and ###$P < 0.0001$ compared to VXL. Each circle represents an individual recipient. **f, g** Kaplan–Meier curves of sublethally irradiated recipients injected with ETP12 (**f**) and ALL8 (**g**), administered with either vehicle or Dynole 34-2, as a single agent or in combination with VXL chemotherapy. Log-rank (Mantel–Cox) test; ***$P < 0.0001$ compared to vehicle; ###$P < 0.0001$ compared to VXL. The period of administration is indicated in light grey, with the number of recipients for each cohort indicated for ETP12 and ALL8 xenograft models.

**Table 1 Sensitivity of primary T-ALL xenograft to Dynole 34-2.**

| ID | Stage | Subtype | LC$_{50}$ Dynole 34-2 (μM) | NOTCH1 | Mutations |
|---|---|---|---|---|---|
| ETP1 | Diagnosis | ETP | 19.9 | | CTCF (SV); DNM2 (p.K557_K558 > K); JAK3 (p.M511I); WT1 (p.R370fs) |
| ETP5 | Diagnosis | ETP | 49.9 | | PTPN11 (A72V) |
| ETP12 | Diagnosis | ETP | 4.26 | | EZH2 (p.E12_spl) (p.W440G); GATA3 (p.A310_A314 > A) (p.R276Q); JAK1 (p.E1012 > EK); SH2B3 (p.V65A) (p.I257T) |
| ETP13 | Diagnosis | ETP | 7.05 | | EED (p.S259F) |
| ETP14 | Diagnosis | ETP | 5.25 | | GATA3 (p.N286T) (p.S271_W275fs); JAK1 (p.S703I); PTEN (SV); RB1 (SV); CDKN2A/B (SV) |
| ALL8 | Relapse | Mature | 5.76 | | ALK (p.E1435del); ASXL1 (p.D863G); BRCA2 (p.C1290Y); C3orf35 (p.A29T); EPHA7 (p.K941Q); FBXW7 (p.R465C); KDM6A (p.A30T); NT5C2 (p.R367Q); SCN5A (p.R481W); SMARCA4 (p.R1189Q) |
| ALL29 | Diagnosis | Mature | 4.51 | 46 | BAI1 (p.Q440R); BCL11B (p.G34fs); EPHB3 (p.A498T); FAT1 (p.G855R); JAK1 (p.M206K); NOTCH1 (p.P2514fs) (p.R1598P); PIK3R5 (p.G612S); SMYD1 (p.A107E) |
| ALL33 | Diagnosis | Mature | 2.43 | 48 | KDM6A (p.VC1110fs); NOTCH1 (p.N325Y); PIK3CD (p.N334K) |

ID identification of the patient-derived xenografts; LC$_{50}$ Dynole 34-2 (μM) median lethal concentration (LC$_{50}$) of Dynole 34-2 for each sample, as assessed in Supplementary Fig. 7a, b; Mutations genetic lesions identified in each sample, by next-generation sequencing, as previously described[21]; NOTCH1 fraction of leukemic blasts harbouring activating mutations of NOTCH1; Stage stage of the disease when sample was harvested; Subtype subtype of T-ALL for each sample; SV structural variant (amino acid change and position) indicated for each gene product.

regulation of growth factor-induced signalling and cell–cell interactions[28–30]. Given the important role of the micro-environment for LSCs[63–65], we postulated that inhibition of Dynamin might impair their stem cell-like properties and sensitize them to chemotherapy. Here, we show that the Dynamin small molecule inhibitor Dynole 34-2 simultaneously blocks multiple signalling pathways in pre-LSCs and LSCs by preventing receptor-mediated endocytosis (Supplementary Fig. 10). Consequently, Dynole 34-2 can overcome the chemoresistance of relapse-inducing cells and improve survival in relevant models of human acute leukemia.

Environmental cues produced by the niche are important for maintaining quiescent relapse-inducing cells and mediating chemoresistance. In the context of acute leukemia, the acquisition of activating mutations of growth factor-induced signalling pathways likely reflects clonal selection for cells that can overcome niche dependence[12,13,43]. Although the merit of targeting these signalling pathways is yet to be fully explored in clinical trials, adaptive changes or selection for clones harbouring activating mutations of growth factor-induced signalling might lead to rapid resistance due to the inherent plasticity of relapse-inducing cells[63]. For example, proteomic profiling of T-ALL samples carrying JAK3 mutations following acute pharmacological inhibition shows upregulation of the MAPK and AKT signalling pathways[27]. In addition to compensatory changes, systemic toxicity of JAK inhibitors (myelosuppression) and Notch1 inhibitors (gastrointestinal) remains problematic[24,36]. In this study, we show that Dynole 34-2 kills leukemic cells by blocking the transduction of multiple growth factor-induced signalling pathways (IL-7, Notch1, IL-3, GM-CSF and SCF) that are essential for the survival of normal and malignant progenitors[66,67]. Accordingly, Dynole 34-2 was very effective in pre-LSCs (Supplementary Fig. 2a) and in vivo treatment led to a 100-fold reduction in pre-LSC numbers in our Lmo2-transgenic model of T-ALL (Fig. 3c). The presence of mutually exclusive activating mutations of either IL-7 or Notch1 signalling pathway in pre-LSCs was not sufficient to limit the efficacy of Dynole 34-2 as a single agent, but conferred resistance to inhibitors exclusively targeting these signalling pathways (Supplementary Fig. 2a, b). Unlike the Ba/F3-IL7R cell line, which relies exclusively on IL-7 for growth and survival, LSCs depend upon several growth factor signals from the microenvironment[11] and thus, the constitutive activation of STAT5 alone was not sufficient to confer resistance to Dynole 34-2 in pre-LSCs to the extent of what we observed with Ruxolitinib

(Supplementary Fig. 2b). Consistent with the decreased sensitivity of Lmo2$^{Tg}$;STAT5-CA$^{Tg}$ and Lmo2$^{Tg}$;N1-ICD$^{Tg}$ thymocytes to Dynamin inhibition (Supplementary Fig. 2a, b), the anti-leukemic activity of Dynole 34-2 as a single agent was limited in overt leukemia, where LSCs have acquired collaborative mutations of multiple signalling pathways (Supplementary Figs. 5a and 6b). Previous studies in different models of acute leukemia also reported that leukemic cells harbouring activating mutations of signalling pathways remained responsive to growth factor stimulation[21,68,69]. Accordingly, LSCs remained responsive to external ligands despite the presence of activating mutations of signalling pathways, allowing Dynole 34-2 to block receptor-mediated endocytosis and downstream signalling triggered by these ligands (Fig. 5c and Supplementary Fig. 6c, d). Importantly, the efficacy of Dynole 34-2 irrespective of activating mutations of growth factor-induced signalling could be explained by the importance of Dynamin for the CXCR4/CXCL12-mediated response to the microenvironment, which promotes survival of LSCs through the activation of the MAPK and AKT signalling pathways in acute leukemia[15,19,70]. Our data support previous reports confirming the efficacy of Dynole 34-2 to inhibit internalization of signalling receptors and downstream signal transduction in different malignancies[35,71]. However, further genetic molecular assays with cytokine-dependent cells lacking Dynamin or expressing specific mutants of Dynamin will be required to characterize the specificity of Dynole 34-2 for targeting Dynamin in the biological context of leukemia. In addition to blocking DDE, Dynole 34-2 may be perturbing vesicle recycling that regulates downstream signal amplification and crosstalk with other pathways[72], which might explain the decreased activation of the mechanistic target of rapamycin complex 1 (mTORC1) pathway observed in Lmo2$^{Tg}$ pre-LSCs (Fig. 2a). Taken together, our data demonstrated the efficacy of Dynole 34-2 at blocking multiple signalling pathways in LSCs, despite the activation of specific growth factor-induced signalling pathways in overt leukemia.

Increased expression of Dynamin is associated with poor outcome and chemoresistance in several malignancies, including T-ALL[73]. Glucocorticoid resistance (GC) is a major hurdle in T-ALL that is frequently driven by activating mutations of growth factor-induced signalling[8,74]. Loss of phosphatase and tensin homolog leading to AKT1 activation can also induce GC resistance[75]. Inhibition of specific signalling pathways using small molecules such as Ruxolitinib reverted GC resistance in leukemic cells[76], suggesting that GC sensitivity can be modulated by

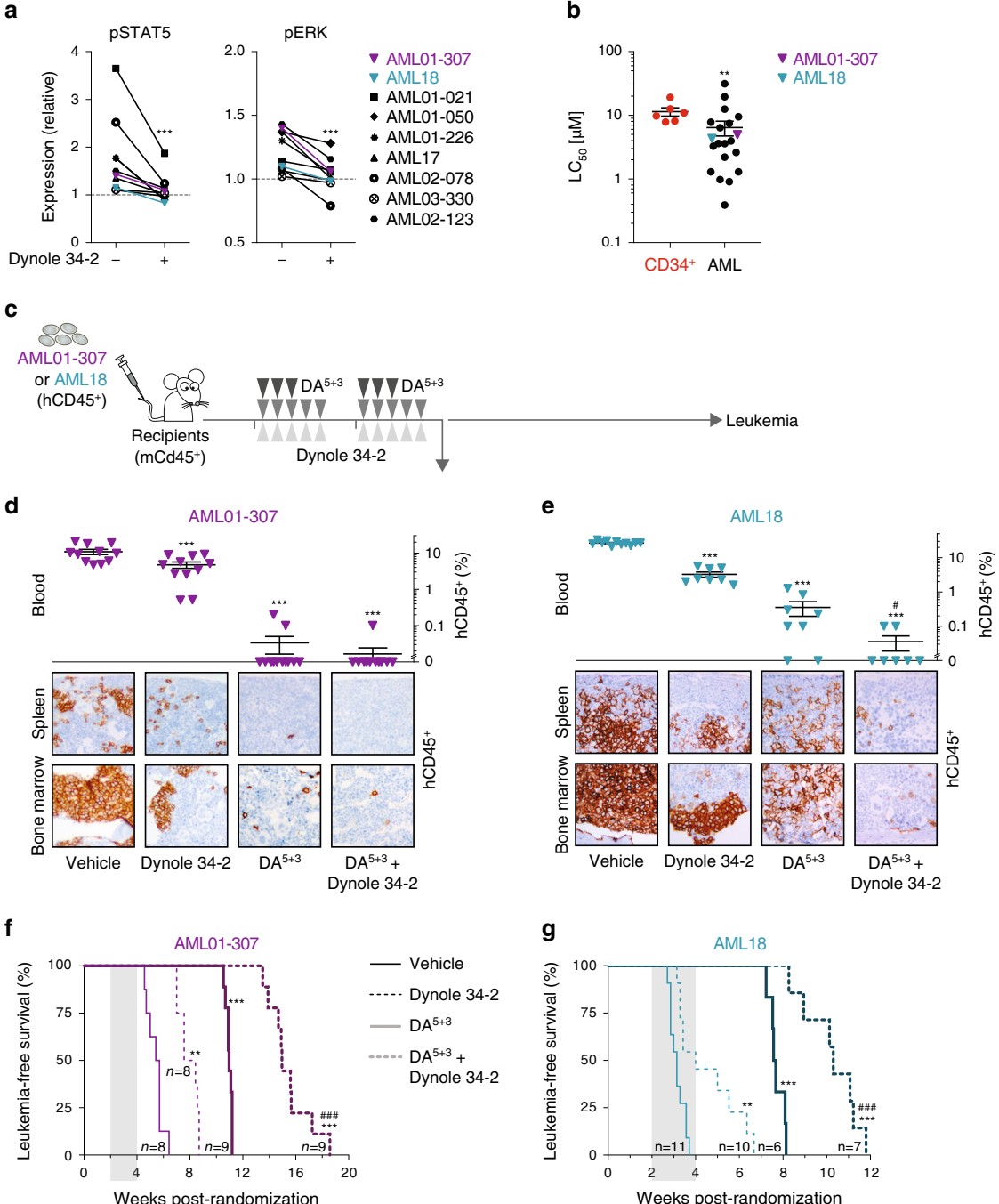

**Fig. 7 Dynole 34-2 exhibits activity in human AML. a** Relative levels of activated STAT5 (pSTAT5, left) and ERK (pERK, right) in patient-derived AML cells treated with vehicle and Dynole 34-2, after in vitro stimulation with IL-3 (left) and SCF (right). AML01-307 is depicted with solid purple triangles, AML18 with solid turquoise triangles, and all other AML samples in black. Levels in unstimulated xenograft cells treated with vehicle were used as control, and reported as 1 (dashed line). Mean ± SD, two-way ANOVA with Bonferroni correction test; **$P < 0.01$ and ***$P < 0.001$ as compared to vehicle ($N = 14$ primary AML, performed in duplicate). **b** Median lethal concentration ($LC_{50}$) of Dynole 34-2 for patient-derived AML cells ($N = 22$ leukemias), assessed by performing clonogenic assays for 10 days (Supplementary Fig. 9c, d). Human CD34-positive HSPCs (red dashed line; CD34+) from healthy donors ($N = 6$) were used as controls. Mean ± SEM, two-tailed Mann–Whitney test; **$P < 0.01$ compared to CD34+ control cells. Each circle represNOTCH1ents an individual human sample. **c**, Experimental setting for testing the efficacy of Dynole 34-2, DA5+3 and combination therapy in xenograft models of human AML. Sublethally irradiated NRGS recipients were randomized after engraftment was confirmed in the peripheral blood and subsequently treated when the average proportion of human leukemic cells in the peripheral blood reached 1%. **d**, **e** Proportion of patient-derived cells (%hCD45+) in the peripheral blood (top), as well as immunochemistry against hCD45+ leukemic cells in the bone marrow and spleen (bottom) of recipients injected with AML01-307 (**d**) and AML18 (**e**), 24 h after the last drug injection. Scale bars: 10 μm. Mean ± SEM, two-way ANOVA with Tukey's correction test; ***$P < 0.0001$ compared to vehicle; #$P < 0.05$ and ###$P < 0.0001$ compared to VXL. Each triangle represents an individual recipient. **f**, **g** Kaplan–Meier curves of sublethally irradiated recipients injected with AML01-307 (**f**) and AML18 (**g**), administered with either vehicle or Dynole 34-2, as a single agent or in combination with DA5+3 chemotherapy. Log-rank (Mantel–Cox) test; ***$P < 0.0001$ compared to vehicle; ###$P < 0.0001$ compared to DA5+3. The period of administration is indicated in light grey, with the number of recipients for each cohort indicated.

**Table 2 Sensitivity of primary AML to Dynole 34-2.**

| ID | Stage | Blast (%) | FLT3-ITD | Karyotype | LC50 Dynole 34-2 (µM) | ASXL1 | EZH2 | PHF6 | TP53 | RUNX1 | GATA2 | FLT3 | JAK2 | KIT | NRAS | PTPN11 | DNMT3A | TET2 | IDH1 | IDH2 | NPM1 | SRSF2 | U2AF1 | BCORL1 | BCOR | Genomic class |
|---|---|---|---|---|---|---|---|---|---|---|---|---|---|---|---|---|---|---|---|---|---|---|---|---|---|---|
| AML01-152 | Relapse | 85 | Negative | +4, +8 del9q | 31.4 | | 17 | | | | | | | | | | | 34 | | | | | | | | Chr Spl |
| AML01-012 | Relapse | 99 | Negative | ND | 19.6 | | | | | | | | | | | | | | | | | | | | | NPM1 |
| AML04-155 | Relapse | 72 | Negative | Complex | 12.5 | 44 | | | 83 | | | | | 48 | | | 44 | | | | 60 | | | | | Com TP53 |
| AML01-086 | De novo | 51 | Negative | +4,+8,+8 | 9.5 | | | | | | | | | | | | | 35 | | | | | | 6 | 2 | No class |
| AML03-124 | Relapse | 45 | Negative | Normal | 7.8 | 34 | | | 12 | 18 | | | | | | | | | | | | 50 | | | | Chr Spl |
| AML01-021 | Relapse | 65 | Negative | Normal | 7.4 | 7 | | | 38 | 26 | | | | | | | | 32 | 24 | | | 34 | | | | Chr Spl |
| AML01-072 | De novo | 86 | Negative | Normal | 6.4 | | | | | | | 64 | 2 | | | | 48 | | | | 60 | | | | | NPM1 |
| AML18 | sAML | NA | 5.22 | t(9;11), MLL | 4.3 | | | | | | | | | 60 | | | | | | | | | | | | No class |
| AML01-064 | De novo | NA | Negative | inv(3) | 3.95 | | | | | | 51 | | | | 60 | | | | | | | | | | | Inv3 |
| AML02-012 | Refractory | 52 | Negative | Monosomal | 3.6 | | | 43 | 50 | 45 | | | | | 36 | | | | | | | | | | | Com TP53 |
| AML01-226 | De novo | 50 | Negative | Normal | 3.5 | | | | | | | 7 | | | | | | | | | | | | 5 | | Chr Spl |
| AML01-050 | De novo | 76 | 3.3 | Normal | 3.3 | | | | | | | | | | | | | | | | | | | | | Com TP53 |
| AML17 | De novo | NA | 86.5 | +8 | 2.6 | | | | | | | 86.5 / 0.05 | | | | | | | | | | | | | | No class |
| AML01-016 | De novo | 33 | NA | Normal | 2.2 | | | | | | | | | 4 | | | | 60 | | | 52 | | | | | NPM1 |
| AML01-318 | Relapse | 73 | Negative | ND | 1.3 | | | | 96 | 44 | | 7.7 | | 28 | 15 | | 46 | | | | 58 | | | | | Com TP53 |
| AML01-044 | De novo | 88 | 7.73 | t(1;3) | 1.3 | | | | | 35 | | 4.6 | | | | | | | | | 18 | | | | | NPM1 |
| AML04-015 | Relapse | 89 | Negative | Normal | 0.98 | | | | | | | | | | | | | 43 | 43 | | | | | 100 | | NPM1 |
| AML03-330 | De novo | 65 | Negative | Normal | 0.91 | | | | | | | | | | | | | | | | | 48 | | | | Chr Spl |
| AML02-123 | De novo | 87 | Negative | idic(21), del (9q) | 0.39 | 50 | | | | | | | | | | | | | | | | | | | | Chr Spl |

Blast (%) proportion of blasts in each sample; *Chr Spl* chromosome splicing; *Com TP53* complex with *TP53* mutation; *FLT3-ITD* detection of the FLT3-ITD mutation, with the fraction of leukemic blasts harbouring FLT3-ITD; *Genomic class:* Genomic class subtype of AML for each sample, classified based on genetic abnormalities; *ID* identification of the patient-derived xenografts; *Inv3* inversion chromosome 3; *Karyotype* chromosomal anomalies observed in each sample; *LC50 Dynole 34-2 (µM)* median lethal concentration (LC50) of Dynole 34-2 for each sample, as assessed in Supplementary Fig. 9c, d; *Mutations* genetic lesions identified in each sample, with the proportion of blasts carrying identified mutations of the listed genes; *Stage* stage of the disease when sample was harvested.

blocking the response of relapse-inducing cells to environmental cues. The ability of Dynole 34-2 to block the activation of STAT5 and AKT may partly explain the synergy observed in T-ALL with the VXL chemotherapy. Moreover, the inhibition of amino acid uptake and the downstream mTORC1 activation by Dynole 34-2 may enhance the effect of L-Asparaginase[77], which kills leukemic cells by rapidly depleting the pool of asparagine in the micro-environment[78]. Using our model of *Lmo2*-transgenic T-ALL, we have shown that Dynole 34-2 enhanced the efficacy of chemotherapy, independently of the presence of collaborative mutations of signalling pathways in the primary leukemias tested, suggesting that multiple fundamental processes regulated by DDE are essential for LSCs. Importantly, we have also shown in different models of acute leukemia that Dynole 34-2 reduced leukemic burden in association with conventional chemotherapy, confirming the importance of DDE for survival of leukemic cells responsible for relapse. Therefore, the enhanced chemosensitivity induced by Dynole 34-2 in both T-ALL and AML may also be related to the importance of Dynamin for actin function in microtubule remodelling, which modulates cytokinesis and DNA repair[39,79]. However, the effects of Dynamin inhibition on nutrients uptake and DNA repair in acute leukemia remain to be investigated.

Genomic studies of paired diagnosis and relapse samples in acute leukemia suggest that recurrence arises from either clonal evolution of pre-LSCs or LSCs present in the diagnostic sample[2,80]. Therefore, models that can identify and directly measure the effects of therapeutics on these cells are paramount for improving cure rates for all leukemias. Here we show that Dynole 34-2 blocks signal transduction and promotes chemo-sensitivity of relapse-inducing cells in diverse and relevant models of acute leukemia that recapitulates the human disease. Importantly, these studies revealed several potential advantages of Dynamin inhibitors over more specific approaches to target relapse-inducing cells. First, given the many known and unknown signals provided by the microenvironment[81], targeting multiple pathways is likely to be more effective and could overcome the rapid resistance-promoting adaptive responses to chemotherapies and targeted therapies[47]. In addition, therapeutic doses of Dynole 34-2 appear well tolerated in combination with chemotherapy, with no obvious myelosuppression or gut toxicity, despite inhibiting multiple signalling pathways that are important for hematopoiesis. In aggregation, our results provide a significant conceptual advance in therapeutic strategies for eradicating relapse-inducing cells by revealing that Dynamin is a druggable target in the most common types of acute leukemia. This study presents preclinical evidence that Dynamin inhibitors are worthy of testing in a broad range of cancers, where signals from the microenvironment promote disease progression and therapeutic resistance of relapse-inducing cells[63].

## Methods

**Retroviral infection of Ba/F3 cells.** The coding sequence of murine *Stat5a* and *Il-7r* was amplified by PCR from cDNA of bone marrow mononuclear cells of a healthy WT donor using primers mStat5a-FL-Fw and mStat5a-FL-NOstop-Rv, and mIl7r-FL-Fw and mIl7r-FL-Rv, respectively (Table 3). The PCR product was cloned into the *XhoI* and *EcoRI* sites of pMIG-R1, which was kindly given by Pear et al.[82,83]. To generate the Stat5-CA mutant[41], targeted mutagenesis of the amino acids His[298]→Arg and Ser[710]→Phe of mStat5a was performed using the primers mStat5a_H298R-Fw, mStat5a_H298R-Rv, mStat5a_S710F-Fw and mStat5a_S710F-Rv (Table 3), and the presence of the mutations was verified by sequencing (Micromon, Monash University, Australia).

The doxycycline-inducible dual-colour TRMPVIR (TRE-dsRed-miR30/shRNA-PGK-Venus-IRES-rtTA3) retroviral vector[84] was kindly given by R. A. Dickins and was modified by replacing the dsRed-miR30/shRNA sequence by a T2A self-cleaving peptide[85] fusioned with the mCherry fluorescent protein (from TRIPZ, Addgene) inserted into the BamHI and EcoRI sites of the plasmid downstream of the TRE promoter to generate the TxTCPVIR (TRE-gene-T2A-mCherry-PGK-Venus-IRES-rtTA3) retroviral vector. Using standard cloning techniques, we

**Table 3 Primers used for cloning of murine genes.**

| Gene | Name of the primer | Sequence |
|---|---|---|
| *Il-7r* | mIl7r-FL-Fw | 5′-ttttagatctagaaggctcgagaccATGATGGCTCTGGGTAGAGCT-3′ |
| | mIl7r-FL-Rv | 5′-tttttttagatctagaagggaattcTTTGTTTTGGTAAAAACTAGACATGGT-3′ |
| *Stat5a* | mStat5a-FL-Fw | 5′-TTTTGAATTCTCAGGACAGGGAGCTTCTAGCGGA-3′ |
| | mStat5a-FL-NOstop-Rv | 5′-ttttgaattcGGACAGGGAGCTTCTAGCGGA-3′ |
| | mStat5a_H298R-Fw | 5′-CGCAGGGCTGAGCGCCTGTGCCAGCAG-3′ |
| | mStat5a_H298R-Rv | 5′-CGCAGGGCTGAGCGCCTGTGCCAGCAG-3′ |
| | mStat5a_S710F-Fw | 5′-TGAGTTCGTCAATGCATTCACAGATGCCGGA-3′ |
| | mStat5a_S710F-Rv | 5′-TCCGCCATCTGTGAATGCATTGACGAACTCA-3′ |
| *mKit* | mKit_FL-F | 5′-TTTTCTCGAGATGAGAGGCGCTCGCGGCGCCTGGGATCT-3′ |
| | mKit_FL-R | 5′-TTTTAAGCTTTCAGGCATCTTCGTGCACGAGCA-3′ |
| *mGmcsfr* | mGmcsfr-FL-F | 5′-TTTTAGATCTATGACGTCATCACATGCCATGAACATCA-3′ |
| | mGmcsfr-FL-R | 5′-TTTTAAGCTTCTAGGGCTGCAGGAGGTCCTTCCTGA-3′ |

*Gene* loci amplified, *Name of the primer* recorded name for each oligonucleotide primer used for cloning; *Sequence* for each primer used.

inserted the full-length sequence of *Stat5-CA* from the pMig-R1 construct into the TxTCPVIR vector to generate the TxTCPVIR-*Stat5-CA* construct. All subcloned constructs were verified by DNA sequencing (Micromon, Monash University, Australia). Detailed cloning strategies, primers sequences and vectors are available on request.

Ba/F3 cells (kindly provided by Ms Sandra Mifsud from The Walter and Eliza Hall Institute of Medical Research) were infected with pMIG-R1/*Il7r* retroviral supernatants produced in 293T cells, and transduced cells were sorted with a BD Influx (BD Australia) and selected by culture in medium supplemented with 5 ng/mL rmIL-7 (#217-17, Pepro Tech, Rocky Hill NJ, USA) for 7 days. Ba/F3 cells stably expressing *Il7r* (Ba/F3-IL7R) were infected with TxTCPVIR-*Stat5-CA* retroviral supernatants produced in 293T. Cell sorting was performed with a BD Influx and transduced cells were treated with doxycycline for 48 h to induce the expression of the *Stat5-CA*-T2A-mCherry fusion protein.

**Dynamin inhibitors**. The synthesis of Dynole 34-2 and Dyngo 4a compounds was performed as previously described[32,33] and was kindly provided by A. McCluskey. For in vitro assays, the specific Dynamin inhibitors Dynole 34-2, Dyngo 4a and Dynasore (#S8047, Selleckchem, Sapphire Bioscience Pty Ltd, Redfern NSW, Australia) were made as stock solutions in 100% dimethyl sulfoxide (DMSO), which were stored frozen. These compounds were diluted in 50% v/v DMSO/20 mM Tris-HCl pH 7.4 immediately prior to use. For in vivo administration, Dynole 34-2 was resuspended in pre-warmed HRC#5 (Pharmatek, San Diego CA, USA) 1 : 9 phosphate-buffered saline (PBS) 1× pH 7.4 at 37 °C, and sonicated 10 times for 30 s using a Bioruptor Plus (Biogenode) immediately prior to injection at 30 mg/kg, twice daily.

**IL-7 receptor studies**. For dynamic analysis of IL-7R internalization, $1 \times 10^5$ Ba/F3-IL7R cells were incubated with Phycoerythrin (PE)-conjugated α-CD127 (A7R34, BioLegend) for 30 min on ice, washed twice in cold PBS 1×, resuspended in serum-free Dulbecco's modified Eagle's medium (DMEM) and stimulated with 50 ng/mL IL-7 at 37 °C for the indicated intervals. Cells were fixed, permeabilized and incubated with either α-clathrin (ab21679, Abcam Australia) or α-Rab5 (ab18211, Abcam Australia), as previously described[14]. Washed cells were incubated in permeabilization buffer with donkey Alexa Fluor 488-conjugated anti-rabbit secondary antibody (A-21206, Molecular Probes, Life Technologies) for 1 h on ice. Cells were washed, incubated with 1 ng/mL 4′,6-diamidino-2-phenylindole for 15 min, washed twice and mounted using Mowiol mounting medium (81381, Sigma-Aldrich Pty Ltd, Castle Hill, NSW). For each independent experiment, at least 50 different fields of view were collected for blinded quantification, for each time point. All images were acquired using a Nikon A1r Plus SI inverted confocal microscope (Nikon Australia, Rhodes, NSW) using a Plan Apo ×60 oil objective.

**Flow cytometry**. Flow cytometry analyses were performed as previously described[10,86] on single-cell suspensions stained using BD Pharmingen antibodies (BD Australia, North Ryde NSW, Australia) against mouse CD4 (RM4-5), CD8 (53-6.7), CD25 (PC61.5), CD44 (IM7), CD45.1 (A20), CD45.2 (104), Thy1.2 (53-2.1; #561616), TCRβ (H57-597), CD117 (ACK45), CD135 (A2F10.1), CD150 (TC15-12F12.2; #562373), B220 (RA3-6B2; #563708), CD3 (145-2C11; #563004), CD19 (ID3; #557655), Gr-1 (RB6-8C5; #563299 and #565033), CD11b (M1/70; #563015 and #560455), Ter119 (Ter119; #565523 and #557909), Sca-1 (D7; #558160) and an eBioscience antibody (eBioscience, Invitrogen) against mouse NOTCH1 (22E5). DN3 populations and IL-7R were stained using BioLegend antibodies (Australian Biosearch, Balcatta WA, Australia) against mouse CD28 (E18) and CD127 (A7R34), respectively. Apoptosis was measured using an antibody against AnnexinV (#556420, BD Australia) and the permeable Sytox Blue (BD Australia) following the manufacturer's protocol.

Flow cytometry analyses on human T-ALL cells were done on single-cell suspensions using BD Pharmingen antibodies against human CD45 (HI30; #557748), NOTCH1 (MHN1-519), and BioLegend antibodies against human KIT (A3C6E2) and CD127 (A019D5). Patient-derived T-ALL xenografts stained for surface markets were then fixed and permeabilized using the BD Cytofix/Cytoperm™ Kit (#554714, BD Australia), and stained using a BD Pharmingen antibody against human cleaved Caspase 3 (C92-605) and a BioLegend antibody against human cytoplasmic CD3 (HCHT1).

Phosphoflow analysis was performed as previously described[14]. Briefly, $2 \times 10^6$ cells stained for surface markers were subsequently resuspended into PBS 1× + 4% v/v paraformaldehyde and incubated for 10 min at 4 °C for fixation. Fixed cells were washed twice using PBS 1×, then resuspended in pre-chilled at −20 °C Perm Buffer III (BD Pharmingen) and incubated 30 min at 4 °C for permeabilization. Cells were washed twice using PBS 1× and incubated overnight at 4 °C in PBS 1× + 2% v/v fetal calf serum (FCS) with Cell Signalling purified antibodies (Genesearch Pty Ltd, Arundel QLD, Australia) against pSTAT5 (Tyr694; #9359), phospho-p38 MAPK (Thr180/Tyr182; #4511), phospho-p44/42 (Erk1/2; #9102), phospho-S6 (Ser235/236; #4856) and phospho-Akt (Ser473; #9271) or isotype control. For intracellular HES1 detection, cells stained for surface markers were subsequently fixed and permeabilized using the BD Cytofix/Cytoperm™ Kit and incubated overnight at 4 °C in PBS 1× + 2% v/v FCS with a purified antibody against HES1 (D6PU; #11988, Cell Signalling) or isotype control. After staining with primary antibodies for phosphoflow or intracellular detection, cells were incubated in permeabilization buffer with donkey Alexa Fluor 488-conjugated anti-rabbit secondary antibody (A-21206, lot #1910751, Molecular Probes, Invitrogen) or goat Alexa Fluor 546-conjugated anti-rabbit secondary antibody (A-11035, lot #1904467, Molecular Probes, Invitrogen) for 1 h on ice, and washed twice in PBS 1×. Fluorescence-activated cell sorting (FACS) analysis was performed using LSRII and LSR Fortessa cytometers, following the gating strategies described in Supplementary Fig. 11. Cell sorting was performed with a FACSAria or BD Influx.

**Western blottings**. Whole-cell lysates were prepared in SLAB buffer (4.5 mM Tris HCL pH 6.8, 1% SDS, 7.2% v/v Glycerol, 2% β-mercaptoethanol), boiled and subjected to electrophoresis on Mini-Protean TGX 10% SDS acrylamide precast gels (Bio-Rad Laboratories Pty Ltd, Gladesville NSW Australia). Proteins were electrotransferred onto a polyvinylidene difluoride membrane (Amersham Biosciences) and probed with Cell Signalling purified antibodies (Genesearch Pty Ltd, Arundel QLD, Australia) against pSTAT5 (Tyr694; #9359), STAT5 (D2O6Y; #9363) or HES1 (D6PU; #11988) followed by goat anti-rabbit or rabbit anti-mouse IgG conjugated to horseradish peroxidase (HRP; EMD Millipore). Loading was assessed by probing the membranes with HRP-conjugated rabbit monoclonal antibody against β-Actin (13E5; #5125, Cell Signalling Technology). Chemiluminescence was detected by using a ChemiDoc MP Imaging System (Bio-Rad Laboratories Pty Ltd, Gladesville NSW Australia).

**Mouse models**. All experiments were pre-approved by the AMREP Animal Ethics Committee. The current study was performed by using the previously described *CD2-Lmo2* (*Lmo2*^Tg) mouse model[87]. Mice cohorts were generated by cross-breeding with the *Lck-Notch1-IC9* (*N1-ICD*^Tg)[45] generously given by T. Hoang[86], the *Lck-STAT5b-CA* (*STAT5-CA*^Tg)[88] kindly given by M.P. McCormack, the *Mx1-Cre* transgenic (*Mx*)[89] and the *ROSA26-YFP* Cre-reporter (*YFP*)[90] mouse strains. All mouse lines were backcrossed onto a C57BL/6J background for ten generations and maintained in pathogen-free conditions according to institutional animal care guidelines.

**Co-culture assays**. Pre-leukemic and leukemic thymocytes were purified by flow cytometry and co-cultured on GFP-positive OP9 and OP9-DL1 stromal cell lines,

as described previously[91]. Thymocytes were co-cultured in reconstituted alpha-MEM medium (#12318, Gibco, Grand Island NY, USA) supplemented with 20% FCS (#12318), HEPES 10 mM (#15630-080), sodium pyruvate 1 mM (#11360-070), β-mercaptoethanol 55 μM (#21985-023), Glutamax 2 mM (#15750-060), 1% Penicillin/Streptomycin (#15140-122), 5 ng/mL rmFLT-3 ligand (#250-31 L, Pepro Tech), 5 ng/mL rmIL-7 (#217-17, Pepro Tech) and 25 ng/mL rmSCF (#250-03, Pepro Tech). After 48 h of co-culture, thymocytes were counted and analysed by FACS.

**Clonogenic assays**. Bone marrow cells collected from 2-month-old WT mice or CD34-positive (CD34+) stem and progenitor bone marrow cells from healthy donors were seeded at a density of $5 \times 10^4$ cells per 35 mm dish in semi-solid agar for GM colony growth, as previously described[92]. Cultures were incubated at 37 °C in 10% $CO_2$ for 10 days in presence of 10 μg/mL rhIL-3 (#200-03, Pepro Tech), 25 μg/mL rhSCF (#300-07, Pepro Tech), 25 μg/mL rhFLT3-L (#300-19, Pepro Tech), rhIL-6 (#200-06, Pepro Tech), 500 nM StemRegenin 1 (SR1; #1227633-49-9, Stem Cell Technologies) and 35 nM UM-171 (#72914, Stem Cell Technologies)[93,94], then stained with acetyl-cholinesterase and counted for the number of colonies.

**Cytokine-mediated signalling activation assays**. For in vitro IL-7 stimulation assays, cells were starved in serum-free DMEM at 37 °C, then incubated with either vehicle or Dynole 34-2 or Ruxolitinib (INCB018424; #S1378, Selleckchem), or DAPT (GSI-IX; #S2215, Selleckchem) resuspended in DMSO for 1 h. Thymocytes were subsequently stimulated with 25 ng/mL IL-7 at 37 °C for 20 min and washed twice using cold PBS 1×. Cells were incubated with 25 ng/mL Rat IgG (Sigma-Aldrich Pty Ltd, Castle Hill NSW, Australia) and with a PE-conjugated antibody for detection of IL-7R (CD127) for 1 h on ice, as described in the FACS analysis section. PE-conjugated isotype was used as negative control. Cells were washed twice, fixed and phosphoflow analysis performed as described in the FACS analysis section. The analysis was performed in triplicate and the mean fluorescence intensity (MFI) observed without stimulation was reported as 100%.

For in vitro NOTCH1 signalling activation assays, cells were seeded on OP9 and OP9-DL1 stromal cell lines as described in the co-culture assays section and incubated overnight with either Dynole 34-2 or Ruxolitinib or DAPT resuspended in DMSO. Harvested cells were washed twice using cold PBS 1× and incubated with 25 ng/mL Rat IgG (Sigma-Aldrich Pty Ltd, Castle Hill NSW, Australia) with a PE-conjugated antibody for detection of NOTCH1 for 1 h on ice, as described in the FACS analysis section. PE-conjugated isotype was used as negative control. Cells were washed twice, fixed and analysed for intracellular HES1 detection by flow cytometry as described in the FACS analysis section. The analysis was performed in triplicate and the MFI observed in thymocytes co-cultured on OP9 cells was reported as 100%.

**Transplantation assays**. Single-cell suspension of murine thymus cells were intravenously injected into sublethally irradiated (650 Rads) isogenic Ly5.1 (Cd45.1) mice. Fold expansion was calculated by dividing the absolute output number of donor-derived DN3a cells harvested at the experimental endpoint by the respective number of DN3a thymocytes injected in each recipient. Limiting dilution transplantations were performed as described[95], by injecting various doses ($10^1$ to $10^6$) of thymocytes per recipient, and mice were scored positive when T-cell lineage reconstitution assessed 6 weeks after transplantation was >1%. Leukemia-propagating cell frequency was calculated by applying Poisson statistics using the ELDA: Extreme Limiting Dilution Analysis software (Walter and Eliza Hall Institute Bioinformatics, Parkville VIC, Australia)[96]. Limiting dilution transplantations were performed independently from any other transplantation assays, using thymocytes from three different $Lmo2^{Tg}$ mice for each cohort. Kaplan–Meier survival and statistical analysis were performed using GraphPad Prism. 6.0 software (GraphPad Software Inc, San Diego CA, USA).

**Pharmacokinetic analysis**. All experiments were pre-approved by the University of Melbourne Animal Ethics Committee (Ethics ID: 0911153.1). Each mouse received a dose of 30 mg/kg of Dynole 34-2. Five different time points were chosen to measure drug level: 15, 30, 60 120 and 240 min after intraperitoneal injection of the recipients. At each time points, treated mice were killed by lethal injection of 0.01 mL/g of Lethobarb® (325 mg pentobarbitone sodium/mL). Following this, blood was collected via intra-cardiac puncture, blood was then allowed to clot at room temperature for 15 min, then centrifuged at $1500 \times g$ for 6 min to separate out the cells and serum. The serum was collected, weighed, snap frozen in dry ice and stored at −80 °C.

Serum concentration of Dynole 34-2 were determined by ultra-high-performance liquid chromatography with a Dinex UltiMate 3000 LC system (Thermo Scientific), and AB SCIEX QTRAP 5500 system (Sciex) with positive ion electrospray ionization and high sensitivity multiple reaction monitoring. Drug-free mouse serum was used as negative controls. Serum blanks, including calibration standard and quality control samples, were prepared by protein precipitation. Selectivity was evaluated by comparing chromatograms of six blank blood samples from six different sources to make sure there were no significant interfering peaks at retention time of analysis. The maximum serum concentration ($C_{max}$) and the time to reach $C_{max}$ ($T_{max}$) were directly obtained from the

experimental data. The time needed for 50% reduction in the serum concentration (terminal half-life; $T_{1/2}$) was calculated from the mean values ($n = 5$) using GraphPad Prism 8.

**Serial transplantation assays**. For primary transplantation, single-cell suspension of $2 \times 10^6$ thymocytes were intravenously injected into sublethally irradiated (650 Rads) isogenic Ly5.1 (Cd45.1) mice. Analysis was performed 4 weeks after transplantation and $5 \times 10^6$ thymocytes harvested from recipients were subsequently injected into sublethally irradiated (650 Rads) Ly5.1 (Cd45.1) recipients for the secondary and tertiary transplantations, as previously described[5].

**Pre-clinical assays with patient-derived xenografts**. Patient-derived xenograft models were established as previously described[97,98]. All experiments were pre-approved by the Human Research Ethics Committee of the University of New South Wales (UNSW), the UNSW Animal Care and Ethics Committee, the Alfred Health Human Ethics Committee and the AMREP Animal Ethics Committee. For transplantation assays, $10^6$ splenocytes harvested from successfully engrafted primary human ETP-ALL or T-ALL xenografts were injected into nonobese diabetic/severe combined immunodeficient (NOD.Cg-$Prkdc^{scid}$, also termed NOD/SCID) or NOD/SCID/$Il2rg^{tm1wjl}$/SzJ (NSG) mice[99]. Moreover, $1.5 \times 10^6$ cells splenocytes harvested from successfully engrafted patient-derived AML xenografts were injected into nonobese diabetic/severe immunodeficient (NOD.Cg-$Rag1^{tm1Mom}Il2rg^{tm1wjl}$/SzJ, also termed NRG) mice expressing human IL-3, human GM-CSF and human SCF from the SGM3 triple transgene (NRGS)[100]. Engraftment into recipients was monitored by FACS analysis of peripheral blood as previously described[21]. Sublethally irradiated NSG or NRGS recipients were randomized after engraftment was confirmed in the peripheral blood, and subsequently treated when the average proportion of human leukemic cells in the peripheral blood reached 1%. Both males and females were used. The in vivo efficacy of Dynole 34-2 was assessed 24 h after the last injection by FACS analysis and immunohistochemistry as described in Supplementary Methods. Kaplan–Meier survival and statistical analyses were performed using GraphPad Prism. 6.0 software (GraphPad Software, Inc.).

Splenocytes from secondary xenografts were processed as previously described[53]. Human leukemia cells were thawed, washed, and adjusted to $2 \times 10^6$ cells/mL in QBSF-60 media (#160-204-101, Quality Biological, Gaithersburg MD, USA) supplemented with Glutamax 2 mM (Gibco), 1% v/v Penicillin/Streptomycin (Gibco) and 20 ng/ml rhFLT3-L (#300-19, Pepro Tech). Cells were washed in alpha-MEM medium (#12318, Gibco, Grand Island NY, USA) supplemented with 0.5% w/v bovine serum albumin (BSA) (Sigma-Aldrich Pty Ltd, Castle Hill NSW, Australia) and incubated for 1 h at 37 °C prior to in vitro stimulation. For IL-7/IL-7R–mediated signalling assays, cells were incubated with either vehicle or Dynole 34-2 for 1 h. Cells were subsequently stimulated with 125 ng/mL rhIL-7 (#200-07, Pepro Tech) at 37 °C for 20 min and washed twice using cold PBS 1×. Flow cytometric analysis of surface IL-7R levels and intracellular levels of pSTAT5 were performed as described in the FACS analysis section. For activation of NOTCH1-mediated signalling, $5 \times 10^5$ rested leukemia cells were seeded on OP9-DL1 stromal cells. Co-culture on OP9 stromal cells was used as a negative control. After 16 h of co-culture, leukemia cells were washed twice using cold PBS 1×. Flow cytometric analysis of NOTCH1 expression and intracellular levels of HES1 were performed as described in the FACS analysis section. The analysis was performed in duplicate and the MFI observed without stimulation was reported as 1.

All bone marrow or peripheral blood samples from patients with AML were collected after informed consent, in accordance with guidelines approved by the Human Research Ethics Committee of Alfred Health. Human AML cells were thawed, washed, and adjusted to $2 \times 10^6$ cells/mL in Iscove's Modified Dulbecco's Media (#12440053, Gibco, Grand Island NY, USA) supplemented with Glutamax 2 mM (Gibco) and 1% v/v Penicillin/Streptomycin (Gibco). Cells were washed in alpha-MEM medium (#12318, Gibco, Grand Island NY, USA) supplemented with 0.5% w/v BSA (Sigma-Aldrich Pty Ltd, Castle Hill NSW, Australia) and incubated for 1 h at 37 °C prior to in vitro stimulation. For cytokine-mediated signalling assays, cells were incubated with vehicle or Dynole 34-2 for 1 h, and subsequently stimulated with either 25 ng/mL rhIL-3 (#200-03, Pepro Tech), 25 ng/mL rhGM-CSF (#300-03, Pepro Tech) or 125 ng/mL rhSCF (#300-07, Pepro Tech) at 37 °C for 20 min. Cells were washed twice using cold PBS 1× and flow cytometry analyses were done on single-cell suspensions using BD Pharmingen antibodies against human CD45 (#557748), IL-3R (CD123, #560087), GM-CSFR alpha chain (CD116, #554532) and BioLegend antibody against human KIT (A3C6E2). For assessing intracellular levels of pSTAT5 and phospho-ERK, flow cytometry analyses were performed in duplicate on patient-derived AML cells as described in the FACS analysis section. The MFI observed without stimulation was reported as 1.

**Immunohistochemistry**. Spleen and sternum from recipients were collected, incubated at least 24 h in 3.7% neutral buffered formalin (pH 7.4), embedded in paraffin and sectioned at 8 μm onto Superforst-Plus slides. Samples were incubated with an anti-human CD45 antibody (Dako, #M0701) in Dako Autostainer Plus solution, then incubated with a sheep anti-mouse IgG conjugated to HRP and processed as per standard protocols using 3,3′ Diaminobenzidine staining (#ab64238, Abcam) for all immunoblots, as previously described[101]. Samples were

imaged and annotated using an Aperio ScanScope and software system (Leica Biosystems).

**Generation of Ba/F3-SGM3R cells**. The coding sequence of murine *Csf2ra* (*gm-csfr*) and *c-Kit* was amplified by PCR from cDNA of bone marrow mononuclear cells of a healthy WT donor using primers mGmcsfr-FL-F and mGmcsfr-FL-R, and mKit-FL-F and mKit-FL-R, respectively (Table 3). The PCR product was cloned into the *XhoI* and *EcoRI* sites of pMIG-R1, which was kindly given by Pear et al.[82,83]. Ba/F3 cells were infected with pMIG-R1/*Gmcsfr* retroviral supernatants produced in 293T cells, transduced cells were sorted with a BD Influx (BD Australia) and selected by culture in medium supplemented with 25 ng/mL rmGm-csf (#305-03, Pepro Tech, Rocky Hill NJ, USA) for 7 days. Finally, Ba/F3 cells stably expressing *Gm-csfr* were infected with pMIG-R1/*Kit* retroviral supernatants produced in 293T cells to generate Ba/F3-SMG3R cells, which were sorted with a BD Influx and maintained in medium supplemented with 5 ng/mL rmIL-3 (#213-13, Pepro Tech, Rocky Hill NJ, USA).

**Reporting summary**. Further information on research design is available in the Nature Research Reporting Summary linked to this article.

## Data availability

All data generated or analysed during this study are included in this article (and its Supplementary Information files). The relevant data that support the findings of this study are available from the corresponding author upon reasonable request. Source data are provided with this paper.

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

## Acknowledgements

We thank Geza Paukovics, Jeanne LeMasurier, Eva Orlowski-Oliver and Magdaline Costa from the AMREP Flow Cytometry Facility, as well as Prue O'Hare, Kylie Spark and Stephanie Jansen from the Alfred Medical Research & Educational Precinct Animal Services (AMREP AS). We also thank Stephen Cody, Iška Carmichael and Chad Johnson from the Monash Micro Imaging platform, as well as Shilpa Bereeka, Nhu-Y Nguyen and Giovanna Pomilio for technical assistance. This work was supported by a Research Fellowship (#700153) from The Terry Fox Foundation (C.S.T.), a grant-in-aid from the Leukaemia Foundation of Australia (C.S.T), a Fellowship from the Australian National Health and Medical Research Council (R.B.L) and a Senior Medical Research Fellowship from the Sylvia and Charles Viertel Foundation (D.J.C).

## Author contributions

C.S.T. designed, supervised, and performed research, coordinated the interactions between the authors, interpreted the results and wrote the manuscript. S.K.C. performed in vivo experiments, analysed data and wrote the manuscript. J.S. and S.E.S. performed in vitro experiments and analysed data. H.M., V.L., J.A.B. and K.E. performed in vivo xenograft studies and analysed data. L.C. and J.M.S. performed experiments with primary leukemia samples. N.C. and A.M. provided reagents. R.B.L. provided biological samples, contributed to design research and supervised the in vivo xenograft studies. P.J.R., S.M.J. and D.J.C. provided reagents, supported research, contributed to research design and wrote the manuscript.

## Competing interests

The authors declare no competing interests.
