## [Peer Review File · Nature Communications]

Reviewers' comments:

Reviewer #1 (Remarks to the Author); expert in LSK and microenvironment:

In this article the authors report on the efficacy of using Dynamin inhibitor Dynole 34-2 to prevent the chemoresistance of T-ALL and AML via inhibiting key signalling pathways.

The use of Dynamin inhibitors provide a conceptual strategy to more efficiently eradicate LSC and prevent relapse. This provide a conceptual advance especially for the treatment of T-ALL-AML. Nevertheless, there are major issues that need to be addressed.

1- The authors have reported previously that the pre-LSC in the LMO2Tg model is restricted to low cycling DN3 subfractions. Nevertheless they show that in Fig 3-4, Dynole 34-2 is also acting on these cells. What is the difference in expression of pSTAT5 and HES1 comparing pre-LSC DN3 low cycling versus DN3 high cycling cells. One will expect that the treatment with Dynole induce decrease in DN3 non-pre-LSC more specifically. Do they definitively decrease the progression to leukemia in these mice treated by Dymole 34-2 alone?

2- The non-toxic effect of Dynole- plus or minus chemo is only shown by phenotype. It is important to evaluate the effect of this drug on normal HSPCs functionally.

3- From Fig 4e, was the combination treatment able to eradicate leukemia progression? What is the effect on survival of these mice?

4- They already show that pre-LSC and ISC are restricted to DN3 fraction. Here again they show the same. What will be more informative will be genetically, whether the treatment with Dynole plus chemo decrease the transformation between pre-LSC to LSC and thus prevent acquisition of mutations?

5- Dynole treatment on leukemic cells by itself seems less effective than in pre-LSC (fig 5e-f). Will be of interest to see whether this is due to overactivation of signalling pathways like Notch activation via mutations or by other mechanisms?

6- The effect on human T-ALL fraction is limited to ex vivo treatment which clearly did not model the effect in vivo. They should at least evaluate whether treatment of the combination have any effect in vivo.

7- The effect of dynole 34-2 on AML is of interest but is really preliminary. Ex vivo, primary AML are difficult to maintain without cytokine stimulation. Thus addition of dynole ex vivo after cytokine deprivation is not a proof that dynole is acting on AML. Similarly ex vivo AML do not form clear CFU colony. Most important is the in vivo assay but only one AML patient was used. Based on the heterogeneity of AML, it is not possible to conclude anything here. Also it is unclear when the treatment was started when leukemia was detected or before? I will suggest focusing on T-ALL instead or increasing the number of AML tested.

Reviewer #2 (Remarks to the Author); expert in T-ALL and mouse models:

In this work Tremblay and colleagues studied to which extent and how dynamin-dependent endocytosis can be targeted in leukemic cells from T-ALL and AML pathologies using small inhibitory molecules. The general idea of the work is that Dynamin inhibition interferes with endocytosis of several cell surface receptors of cytokines/ligands, that are important extrinsic factors in leukaemia development/propagation. The endocytosis process is indeed crucial for the cytokines/ligands (IL7, DLL, IL3, SCF...) efficient leukemic support of cell growth. Dynamin inhibition is further tested in tandem with commonly used chemotherapy to induce leukemic cell-death, especially in the Leukaemia Stem Cell (LSC) compartment. The paper is basically divided into two parts: a T-ALL section where the proof of concept of the therapeutic strategy is made, especially targeting the LSC compartment, and a AML section, more reduced in terms of data size, that provides the idea that inhibiting Dynamin may also be efficient in other acute leukaemia (albeit only one AML sample is tested in vivo in immune-deficient mice). The results presented are numerous and represent an important piece of data that will be of interest for the wide audience of

Nature Communications. Several functional models are utilized, such as cell lines, mouse models and xenografted samples from human leukaemia patients. The experiments are well- and cleverly-executed providing strong and convincing conclusions, that may open to interesting and novel therapies for acute leukaemia in patients. It is important to stress that this work proposes to target a ubiquitous rather than a specific mechanism to treat different types of acute leukaemia, that in my opinion is a very original idea but that may induce secondary effects.

I have recommendations that will help improving further the manuscript.

1- It is important to explain why DN3a thymocytes from normal thymus are 5 times less sensitive to Dynol 34-2 than pre-LSC DN3a LMO2tg cells. Also, it is not clear if and to which extent Dynamin inhibition alters normal thymopoiesis and BM hematopoiesis. Treatments of WT normal mice with these molecules and a follow up on the blood/thymus/BM parameters should be shown.

2- Have the authors checked whether the DDE inhibition is active on the CXCR4-CXCL12 axis that was shown to be crucial for T-ALL survival and migration? CXCR4 is endocytosed upon CXCL12 ligation and may also be targeted by Dynamin inhibitors.

3- Please specify what means "biological replicates" in every figure where it is used. Not sure what that means and not sure "independent experiments" have been done. The authors should be more precise in the figure legends about the reproducibility of their observations.

4- In many figures the mean values are shown instead of the median value of several replicates. This should also be corrected, as wrong interpretation may be commented thereafter. For instance Figure 2a, Fig S3a, Fig S5e ..etc..

5- Fig2-(a): how MFI are calculated when there are several pics? P-S6? P-p38? Are they truly different? (b) Please use a non-parametric statistic test and not the Student-t test. (c) HES1 protein levels are not so much enhanced upon Notch activation or inhibition- please show results with several Notch targets using transcript analyses.

6- Please explain why (page 7, lanes 7-8) "...LSC carrying activating mutations of IL7R or Notch1 signalling pathways may also be resistant". Will all mutations drive resistance? At the same levels? And to which treatments? Not clearly explained. Especially that page 9/FigS5a-b: activating mutations do not increase resistance.

7- Fig3-(a) please indicate the normal numbers of DN3a cells from WT mice using a dotted line. (c) are these the results shown also in FigS3b, 1ry transplant? If yes, it should be mentioned. Not sure what "Fold expansion" means: compared to injected numbers or to WT DN3a cell numbers?

8- Fig4(b) The authors should show the absolute numbers of DN1, DN2 and DN4 to be able to discuss about the selective/preferential killing of DN3a by Dynol 34-2.

9- Fig5(b) Please explain why DN1/LMO2tg cells only marginally give rise to leukaemia in mice. Because they precede DN3a during T cell development, logically they should be also efficient in promoting leukaemia. (e) Please change the order of treatments on the graph to +/-, then -/+ and +/+. Best would be to show absolute numbers (shown in FigS5e) in the main figure rather than % of cells.

10- Fig6 (d-f) and (e-g): what's the level of IL7R and Notch1 expression in the leukemic cells after they relapse from treatments? Did the cells adapt and changed their cell surface expression of DDE-targeted receptors? Are there any acquisition/selection of novel genetic events compared to untreated cells? Are relapse cells different from the original injected ones? In other words: why are those mice relapsing (especially in Fig6d and e: VXL+Dynole 34-2 seems to be such a very efficient treatment)?

11- Suppl Fig6b: please indicate which type of T-ALL were studied: cortical? Mature?...

12- Showing additional experiments with AML samples will allow to know whether these results are true for several leukaemia patients.

Minor:

1- The AML section is less documented than the T-ALL part. One may propose to take this part out and publish it separately, as a specific story.

2- If one consider treatments in patients, and the use of Dynamin inhibitors in such settings, it would be interesting to start treating mice with the VXL drug combination, and when leukaemia burden has seriously dropped down, treat with Dynamin inhibitors (with or without a combination

of VXL, maybe with even lower doses) in a situation of relapse/refractory disease.
3- T-ALL and AML patient samples should be better described.

Reviewer #3 (Remarks to the Author); expert in endocytosis and signaling:

This work investigates the importance of Dynamin activity and Dynamin-dependent processes in the survival of Leukemic stem cells. To this end, the treatment of choice is the use of various Dynamin inhibitors and a specifically of Dynole 34-2. Using this strategy, the authors showed that Dynole 34-2 is as effective as Ruxolitinib or DAPT in inhibiting DN3a thymocytes, it further impairs Ba/F3 cell growth without impacting on cytokines. Notably, Dynole 34-2 impairs stat5 signaling and its inhibitory effect is relieved in Ba/F3 cells that express a constitutive active Stat5A. These effects correlate with a reduction of cell surface levels of IL7R and Notch1. Notably, in vivo in Lmo2Tg mice, Dynole 34-2, despite its short half-life, reduces the number of thymocytes, which display a large reduction in self-renewal. In these mice, Dynole 34-2 leads to exhaustion of pre-LCS cells. In patient-derived ETP-ALL and mature T-ALL xenografts, Dynole34-2 decrease Sta5a and HES1 and leads to the killing of the leukemic cells. Similarly, in AML models Dynole 34-2 impairs signaling, causes cell death: effects that correlate with the inhibition of endocytosis of multiple receptors, such as c-Kit, GM-CSFR, and IL3R. In conclusion, this work provides convincing evidence that Dynole 34-2 impairs the signaling and internalization of various receptors impacting on the viability of both T-ALL and AML cells.

The in vitro, ex vivo and in vivo evidence of the efficacy of Dynole 34-2 on T-ALL and AML re well documented and corroborated. Dynole 34-2 appears to impact on these leukemic cells possibly by impairing the endocytosis of multiple receptors and in so doing of multiple pathways. These findings are relevant and interestingly.

However, both conceptually and mechanistically, this work is not particularly novel and would seem more appropriate for a specialized pharmacological journal.

This notwithstanding, it would seem necessary to prove that Dynole 34-2 acts specifically through Dynamins (and Dynamin 2 specifically as implied) using CRISPR or knock down or Knock out leukemic T-ALL and AML cells. Also, the assays to measure endocytosis are superficial and limited to the determination of the surface levels of various receptors. Finally, the authors argue that Dynamin 2 might be the main actor in this context, but the evidence in support of this contention is lacking.

May 15, 2020

Dear reviewers,

We appreciate the constructive comments you provided in regards to our manuscript entitled “Small molecule inhibition of Dynamin-dependent endocytosis targets multiple niche signals and impairs leukemia stem cells” for publication in *Nature Communications*. We addressed all of the reviewers’ comments in a point-by-point response attached to this letter, including the suggested changes that are highlighted in the annotated manuscript.

Dr Cedric Tremblay

CONFLICT OF INTEREST: The authors declared no conflicts of interest.

Reviewers' comments:

Reviewer #1 (Remarks to the Author); expert in LSK and microenvironment:

In this article the authors report on the efficacy of using Dynamin inhibitor Dynole 34-2 to prevent the chemoresistance of T-ALL and AML via inhibiting key signalling pathways. The use of Dynamin inhibitors provide a conceptual strategy to more efficiently eradicate LSC and prevent relapse. This provide a conceptual advance especially for the treatment of T-ALL-AML.

Nevertheless, there are major issues that need to be addressed.

Comment 1 - The authors have reported previously that the pre-LSC in the *Lmo2*^{Tg} model is restricted to low cycling DN3 subfractions. Nevertheless they show that in Fig 3-4, Dynole 34-2 is also acting on these cells. What is the difference in expression of pSTAT5 and HES1 comparing pre-LSC DN3 low cycling versus DN3 high cycling cells.

Response: This has been added to the manuscript as follows: We have previously reported that long-term self-renewal and therapeutic resistance is limited to a rare population of cell cycle-restricted pre-LSCs (Tremblay, *et al.* Nature Communications 2018). Using our *TetOP-H2B-GFP*^{KI/+}; *Lmo2*^{Tg} (*H2B-GFP*; *Lmo2*^{Tg}) mouse model, we showed that a small fraction of DN3a thymocytes ($1.7 \pm 0.2\%$, Fig. S3c) retained GFP expression (GFP^{hi}) 2 weeks after withdrawal of doxycycline following a 6-week labelling period, as previously established (Tremblay, *et al.* Nature Communications 2018). In accordance with published gene expression data from GFP^{hi} and GFP^{lo} DN3a thymocytes (Tremblay, *et al.* Nature Communications 2018), levels of pStat5 and Hes1 were not affected by cell cycle kinetics (Fig. S3e), although significantly decreased in DN3a cells from treated mice.

One will expect that the treatment with Dynole induce decrease in DN3 non-pre-LSC more specifically. Do they definitively decrease the progression to leukemia in these mice treated by Dynole 34-2 alone?

Response: This has been added to the manuscript as follows: In this model, administration of Dynole 34-2 decreased the number of DN3a GFP^{hi} cells by 10 fold (Fig. S3d), although cell cycle-restricted DN3a cells did not display increased resistance to Dynole 34-2, as the proportion of GFP^{hi} cells remained the same in treated mice (Fig. S3c). Importantly, Dynole 34-2 significantly impaired repopulation activity and leukemogenicity of GFP^{hi} DN3a thymocytes (Fig. S3f-h). Altogether, this data confirms that Dynole 34-2 has significant *in vivo* activity on long-term self-renewing pre-LSCs as a single agent.

Comment 2 - The non-toxic effect of Dynole- plus or minus chemo is only shown by phenotype. It is important to evaluate the effect of this drug on normal HSPCs functionally.

Response: We agree, and performed competitive transplantations of normal HSPCs. This has been added to the manuscript as follows: We then assessed the safety of the combination of Dynole 34-2 and chemotherapy in non-tumor-bearing mice, as the effect of these drugs on normal hematopoiesis have not been reported. As a single agent, Dynole 34-2 had no detrimental effect on differentiated cells in the thymus or the bone marrow of treated mice (Fig. S5a-c), or the numbers of phenotypic bone marrow stem and progenitor cells (Fig. S5d). On the other hand, chemotherapy significantly decreased the absolute numbers of all progenitor and differentiated cell populations analysed (Fig. S5a-d). To functionally assess the effect of these drugs on stem cell activity, we performed a competitive transplant with total bone marrow

cells from treated mice (Fig. S5e). Unlike chemotherapy, which significantly impairs HSC activity^{47, 48, 49}, Dynole 34-2 had no detrimental effect on HSC fitness and differentiation potential (Fig. S5f-h). Importantly, Dynole 34-2 prevented the chemotherapy-induced decrease of HSC activity and restored normal differentiation in recipients, suggesting a protective effect of Dynole 34-2 in genotoxic stress conditions. Thus, Dynole 34-2 overcomes chemoresistance of pre-LSCs without detrimental effects on normal hematopoietic stem cells and progenitors.

Comment 3 - From Fig 4e, was the combination treatment able to eradicate leukemia progression? What is the effect on survival of these mice?

Response: This has been added to the manuscript as follows: Although chemotherapy did not affect the leukemogenic potential of pre-LSCs, we observed delayed progression to leukemia in recipients injected with thymocytes from *Lmo2*^{Tg} treated with Dynole 34-2 (Fig. S4d). The synergy between Dynole 34-2 and chemotherapy was even more marked on leukemogenicity of treated pre-LSCs in transplantation assays as 80% of recipients remained leukemia-free for up to 6 months (Fig. S4d).

Comment 4 - They already show that pre-LSC and LSC are restricted to DN3 fraction. Here again they show the same. What will be more informative will be genetically, whether the treatment with Dynole plus chemo decrease the transformation between pre-LSC to LSC and thus prevent acquisition of mutations?

Response: As mentioned by this reviewer, we have previously shown that pre-LSC activity is restricted to the DN3 fraction (McCormack, *et al.* Science 2010). Although this author has previously shown that LSCs are enriched within the DN3 fraction in T-ALL using *SCL-LMO1* transgenic mice (Tremblay, *et al.* Genes & Development 2010), this remained to be confirmed in our *Lmo2*^{Tg} model. Given that self-renewal enables pre-LSCs to accumulate additional genetic events promoting clonal selection and progression to leukemia, we postulated that Dynole 34-2 and chemotherapy would affect clonality of pre-LSCs and their capacity of generating T-ALL. Consistent with this idea, there was reduced monoclonality in DN3a cells from *Lmo2*^{Tg} treated with Dynole 34-2 or chemotherapy (Fig. S4c), as assessed by *Tcrβ* rearrangement. Strikingly, the combination therapy eradicated pre-leukemic clones in DN3a cells (Fig. S4c).

Comment 5 - Dynole treatment on leukemic cells by itself seems less effective than in pre-LSC (fig 5e-f). Will be of interest to see whether this is due to overactivation of signalling pathways like Notch activation via mutations or by over mechanisms?

Response: We have performed survival assays with *Lmo2*-transgenic DN3a cells expressing either a constitutively activated form of STAT5 (*STAT5-CA*^{Tg}) or a hyperactive allele of Notch1 (*N1-ICD*^{Tg}). Our data revealed that Dynole 34-2 was less effective in pre-LSCs with overactivation of signalling pathways, suggesting that activating mutations of signalling pathways found in overt leukemia might explain the increased resistance of leukemic cells to Dynole 34-2 treatment. This has been included in the Discussion section.

Comment 6 - The effect on human T-ALL fraction is limited to *ex vivo* treatment which clearly did not model the effect *in vivo*. They should at least evaluate whether treatment of the combination have any effect *in vivo*.

Response: We have shown *in vivo* data using a patient-derived ETP-ALL and mature T-ALL xenograft that were injected into sublethally-irradiated immunocompromised animals (Figure 6c-g). This has been clarified in the Results section.

Comment 7 - The effect of Dynole 34-2 on AML is of interest but is really preliminary. *Ex vivo*,

primary AML are difficult to maintain without cytokine stimulation. Thus adding Dynole 34-2 *ex vivo* after cytokine deprivation is not a proof that Dynole 34-2 is acting on AML. Similarly *ex vivo* AML do not form clear CFU colony.

Response: Patient-derived xenograft cells have been maintained in media supplemented with IL-3, SCF, IL-6 and FLT3-ligand. For *in vitro* stimulation assays (Fig. 7 and S9), AML cells were starved overnight and pre-treated with Dynole 34-2 prior to stimulated with the indicated cytokines. Clonogenic assays with patient-derived AML cells cultured in presence of the relevant cytokines is the gold-standard for assessing drug efficacy. For clarification, cytokines used in clonogenic assays have been mentioned in the manuscript and thoroughly described the Supplementary Methods section.

Most important is the *in vivo* assay but only one AML patient was used. Based on the heterogeneity of AML, it is not possible to conclude anything here. Also it is unclear when the treatment was started when leukemia was detected or before? I will suggest focusing on T-ALL instead or increasing the number of AML tested.

Response: We have generated *in vivo* data using a patient-derived AML xenograft harbouring FLT3-ITD, the most common mutation found in AML. To support this data, we have tested the pre-clinical potential of Dynole 34-2 in combination with chemotherapy using another AML patient-derived xenograft carrying *KRAS*^{G12A} mutant, one of the most common activating mutation of signalling pathway in human AML. As previously described in the Figure legends (Fig. 6 and 7), 'Sublethally-irradiated NRGS recipients were randomized after engraftment was confirmed in the peripheral blood, and subsequently treated when the average proportion of human leukemic cells in the peripheral blood reached 1%.'. For clarification, this information has been added to the Material and Methods section.

Reviewer #2 (Remarks to the Author); expert in T-ALL and mouse models:

In this work Tremblay and colleagues studied to which extent and how dynamin-dependent endocytosis can be targeted in leukemic cells from T-ALL and AML pathologies using small inhibitory molecules. The general idea of the work is that Dynamin inhibition interferes with endocytosis of several cell surface receptors of cytokines/ligands, that are important extrinsic factors in leukaemia development/propagation. The endocytosis process is indeed crucial for the cytokines/ligands (IL7, DLL, IL3, SCF...) efficient leukemic support of cell growth. Dynamin inhibition is further tested in tandem with commonly used chemotherapy to induce leukemic cell-death, especially in the Leukaemia Stem Cell (LSC) compartment. The paper is basically divided into two parts: a T-ALL section where the proof of concept of the therapeutic strategy is made, especially targeting the LSC compartment, and a AML section, more reduced in terms of data size, that provides the idea that inhibiting Dynamin may also be efficient in other acute leukaemia (albeit only one AML sample is tested *in vivo* in immune-deficient mice). The results presented are numerous and represent an important piece of data that will be of interest for the wide audience of Nature Communications. Several functional models are utilized, such as cell lines, mouse models and xenografted samples from human leukaemia patients. The experiments are well- and cleverly-executed providing strong and convincing conclusions, that may open to interesting and novel therapies for acute leukaemia in patients.

It is important to stress that this work proposes to target a ubiquitous rather than a specific mechanism to treat different types of acute leukaemia, that in my opinion is a very original idea but that may induce secondary effects.

Response: We thank this reviewer for highlighting the importance and novelty of this research project. This is very much appreciated.

I have recommendations that will help improving further the manuscript.

Comment 1 - It is important to explain why DN3a thymocytes from normal thymus are 5 times less sensitive to Dynole 34-2 than pre-LSC DN3a *Lmo2*^{Tg} cells.

Response: We agree, and a potential explanation has been added to the manuscript as follows: This decreased sensitivity of wild-type DN3a cells to Dynole 34-2 could be explained by the enhanced survival associated with increased levels of pStat5 as compared to *Lmo2*^{Tg} thymocytes (Shields *et al.* Leukemia 2015; Tremblay *et al.* Leukemia 2016).

Also, it is not clear if and to which extent Dynamin inhibition alters normal thymopoiesis and BM hematopoiesis. Treatments of WT normal mice with these molecules and a follow up on the blood/thymus/BM parameters should be shown.

Response: We now provide more detailed analysis of the effects of Dynole 34-2 on normal hematopoiesis, as well as competitive transplant experiments showing that Dynole 34-2 does not impair HSC function, but prevents chemotherapy-induced detrimental effects of HSC activity.

Comment 2 - Have the authors checked whether the DDE inhibition is active on the CXCR4-CXCL12 axis that was shown to be crucial for T-ALL survival and migration? CXCR4 is endocytosed upon CXCL12 ligation and may also be targeted by Dynamin inhibitors.

Response: Although our studies focused on signalling pathways that are mutated in acute leukemia, we agree that the CXCR4/CXCL12 signalling pathway may in part contribute to the effect observed with Dynole 34-2 and thus, we have included this in the Discussion.

Comment 3 - Please specify what means “biological replicates” in every figure where it is used. Not sure what that means and not sure “independent experiments” have been done. The authors should be more precise in the figure legends about the reproducibility of their observations.

Response: We have defined ‘biological replicates’ for each experiment and indicated how many independent experiments have been performed in the Figure legends.

Comment 4 - In many figures the mean values are shown instead of the median value of several replicates. This should also be corrected, as wrong interpretation may be commented thereafter. For instance Figure 2a, Fig S3a, Fig S5e, etc..

Response: As per transplantation analysis of normal HSCs, fold expansion of pre-LSCs/LSCs are compared using mean values, whereas median values are used for survival analyses.

Comment 5 - Fig2-(a): how MFI are calculated when there are several peaks? P-S6? P-p38? Are they truly different?

(b) Please use a non-parametric statistic test and not the Student-*t* test.

Response: As recommended by experts in the field of flow cytometry, the MFI was calculated using the mean when expression levels follow a normal distribution, and the geometric mean when the distribution is not normal. Using a Mann-Whitney test, the levels of p-S6 are significantly different, whereas the levels of p-p38 were not significantly affected, by Dynole 34-2. We have updated this information in Figure 2.

(c) HES1 protein levels are not so much enhanced upon Notch activation or inhibition- please show results with several Notch targets using transcript analyses.

Response: Given that HES1 expression is oscillatory due to its repressive activity on its own promoter, the increased expression observed following a 10hr co-culture is expected. This experiment has been performed in accordance with recommendations from the team of Prof Warren Pear, an expert in the field (Gerhardt, *et al.* Genes & Dev 2014).

Comment 6 - Please explain why (page 7, lanes 7-8) "...LSC carrying activating mutations of IL7R or Notch1 signalling pathways may also be resistant". Will all mutations drive resistance? At the same levels? And to which treatments? Not clearly explained. Especially that page 9/FigS5a-b: activating mutations do not increase resistance.

Response: We agree, and clarified this by mentioning that the presence of mutually exclusive activating mutations of either IL-7 or Notch1 signalling pathway in pre-LSCs is not sufficient to limit the efficacy of Dynole 34-2 as a single agent, but confers resistance to inhibitors exclusively targeting these signalling pathways (i.e., Ruxolitinib or DAPT). On the other hand, the sensitivity of LSCs to Dynole 34-2 was limited by the presence of collaborative mutations of multiple signalling pathways found in overt leukemia, without preventing the antileukemic activity of Dynole 34-2 as a single agent. Moreover, Dynole 34-2 significantly enhanced the efficacy of chemotherapy, independently of the presence of collaborative mutations of multiple signalling pathways in the primary leukemias tested, suggesting that Dynamine inhibitors effectively target relapse-inducing cells in multiple models of acute leukemia. This has been included in the Discussion.

Comment 7 - Fig3-(a) please indicate the normal numbers of DN3a cells from WT mice using a dotted line.

Response: Data included in Fig. 3a, with description in the Figure Legend.

(c) are these the results shown also in FigS3b, 1ry transplant? If yes, it should be mentioned.

Response: Data presented in Fig. 3c and S3b are from independent experiments. This has been clarified in the Figure Legend, Supplementary Figures and Supplementary Methods sections.

Not sure what "Fold expansion" means: compared to injected numbers or to WT DN3a cell numbers?

Response: Fold expansion was calculated by dividing the absolute output number of donor-derived DN3a cells harvested at the experimental endpoint by the respective number of DN3a thymocytes injected in each recipient. This has been included in the Experimental Procedures section of the revised manuscript.

Comment 8 - Fig4(b) The authors should show the absolute numbers of DN1, DN2 and DN4 to be able to discuss about the selective/preferential killing of DN3a by Dynole 34-2.

Response: The absolute numbers of DN1, DN2 and DN4 thymocyte populations have been included in Fig. S4. From the data we can conclude that although treatment with Dynole 34-2 affects the survival of different thymocyte populations, it preferentially kills DN3a cells (64% decrease, as compared to 56% and 45% for DN1 and DN2, as well as 134% increase for DN4 cells). The same is observed when combined with VXL, with a 75% decrease in DN3a thymocytes as compared to VXL alone, suggesting that Dynole 34-2 selectively sensitizes pre-LSC-enriched DN3a thymocytes to chemotherapy.

Comment 9 - Fig5(b) Please explain why DN1/*Lmo2*^{Tg} cells only marginally give rise to

leukaemia in mice. Because they precede DN3a during T-cell development, logically they should be also efficient in promoting leukaemia.

Response: Although DN1 and DN2 cells precede DN3a during T-cell development, previous studies by our group and others showed that self-renewal and leukemogenicity are restricted to the DN3/DN3a population during the pre-leukemic stage, as well as in overt T-ALL in different murine models (McCormack *et al.* Science 2010, Tremblay *et al.* Genes & Dev 2010, Gerby *et al.* PLoS Genet 2014).

(e) Please change the order of treatments on the graph to +/-, then -/+ and +/+. Best would be to show absolute numbers (shown in FigS5e) in the main figure rather than % of cells.

Response: The order of treatments have been changed and the absolute numbers have been included in Fig. 5, following this reviewer's recommendation.

Comment 10 - Fig6 (d-f) and (e-g): what's the level of IL7R and Notch1 expression in the leukemic cells after they relapse from treatments? Did the cells adapt and changed their cell surface expression of DDE-targeted receptors?

Response: In accordance with our pharmacokinetic analyses of serum levels of Dynole 34-2 in treated mice, the effects of the drug are transient, so the levels of IL-7R and Notch1 were not significantly different than baseline in relapsed cells.

Are there any acquisition/selection of novel genetic events compared to untreated cells? Are relapse cells different from the original injected ones? In other words: why are those mice relapsing (especially in Fig6d and e: VXL+Dynole 34-2 seems to be such a very efficient treatment)?

Response: We agree with this reviewer, regarding this critical question: how conventional chemotherapy and other drugs like Dynole 34-2 affect clonal evolution following treatment. However, this question is beyond the scope of the current study, which focuses on the pre-clinical efficacy of Dynole 34-2 as a novel combination therapy for acute leukemia. Although the short-term treatment of PDX with combination therapy induced a significant loss of leukemic burden in recipient mice, relapse can be explained by the remaining leukemic cells observed in all hematopoietic organs after treatment.

Comment 11 - Suppl Fig6b: please indicate which type of T-ALL were studied: cortical? Mature?...

Response: The subtype of the different T-ALL patient samples tested have been included in Table 1.

Comment 12 - Showing additional experiments with AML samples will allow to know whether these results are true for several leukaemia patients.

Response: We have now performed clonogenic assays with a total of 20 patient-derived AML cells for assessing drug efficacy. Moreover, we have generated *in vivo* data using another patient-derived AML xenograft harbouring *KRAS*^{G12A}, one of the most common activating mutation of signalling pathway in human AML.

Minor:

1 - The AML section is less documented than the T-ALL part. One may propose to take this part out and publish it separately, as a specific story.

Response: We have performed more *in vitro* and *in vivo* experiments using patient-derived AML samples to confirm the efficacy of Dynole 34-2 in AML. Although we agree that the AML

part could be published separately, this would limit the scope of the current study, which aims to demonstrate the efficacy of Dynole 34-2 in different types of hematological malignancies.

2 - If one consider treatments in patients, and the use of Dynamin inhibitors in such settings, it would be interesting to start treating mice with the VXL drug combination, and when leukaemia burden has seriously dropped down, treat with Dynamin inhibitors (with or without a combination of VXL, maybe with even lower doses) in a situation of relapse/refractory disease.

Response: We agree that assessing the efficacy of Dynole 34-2 in relapse/refractory leukemia is an important question to be answered in the clinical context. However, this question is beyond the scope of the current study, which revealed that Dynole 34-2 targeted relapse-inducing cells by blocking multiple signalling pathways that are essential for survival and chemoresistance in acute leukemia. Given that these signalling pathways modulate the response of relapse-inducing cells to chemotherapy, we chose to administer Dynole 34-2 in combination with chemotherapy, for confirming that inhibition of these signalling pathways actually sensitized relapse-inducing cells to chemotherapy.

3 - T-ALL and AML patient samples should be better described.

Response: Genetic information regarding the different T-ALL and AML patient samples have been provided in Tables 1 and 2, respectively.

Reviewer #3 (Remarks to the Author); expert in endocytosis and signaling:

This work investigates the importance of Dynamin activity and Dynamin-dependent processes in the survival of Leukemic stem cells. To this end, the treatment of choice is the use of various Dynamin inhibitors and a specifically of Dynole 34-2. Using this strategy, the authors showed that Dynole 34-2 is as effective as Ruxolitinib or DAPT in inhibiting DN3a thymocytes, it further impairs Ba/F3 cell growth without impacting on cytokines. Notably, Dynole 34-2 impairs stat5 signaling and its inhibitory effect is relieved in Ba/F3 cells that express a constitutive active Stat5A. These effects correlate with a reduction of cell surface levels of IL7R and Notch1. Notably, *in vivo* in *Lmo2^{Tg}* mice, Dynole 34-2, despite its short half-life, reduces the number of thymocytes, which display a large reduction in self-renewal. In these mice, Dynole 34-2 leads to exhaustion of pre-LCS cells. In patient-derived ETP-ALL and mature T-ALL xenografts, Dynole 34-2 decreases Stat5 and HES1 and leads to the killing of the leukemic cells. Similarly, in AML models Dynole 34-2 impairs signaling, causes cell death: effects that correlate with the inhibition of endocytosis of multiple receptors, such as c-Kit, GM-CSFR, and IL3R. In conclusion, this work provides convincing evidence that Dynole 34-2 impairs the signaling and internalization of various receptors impacting on the viability of both T-ALL and AML cells.

Response: We thank this reviewer for the positive and constructive comments.

The *in vitro*, *ex vivo* and *in vivo* evidence of the efficacy of Dynole 34-2 on T-ALL and AML are well documented and corroborated. Dynole 34-2 appears to impact on these leukemic cells possibly by impairing the endocytosis of multiple receptors and in so doing of multiple pathways. These findings are relevant and interestingly. However, both conceptually and mechanistically, this work is not particularly novel and would seem more appropriate for a specialized pharmacological journal.

Response: We thank this reviewer for the constructive comments. We think that our findings provide a significant conceptual advance in therapeutic strategies for acute leukemia that may

be applicable to other malignancies in which signals from the niche are involved in disease progression and chemoresistance.

This notwithstanding, it would seem necessary to prove that Dynole 34-2 acts specifically through Dynamins (and Dynamin 2 specifically as implied) using CRISPR or knock down or Knock out leukemic T-ALL and AML cells.

Response: The specificity and efficacy of Dynole 34-2 for targeting Dynamins have been previously demonstrated (Hill *et al.* J. Med. Chem. 2009; Gordon *et al.* J. Med. Chem. 2013), including studies comparing the efficacy of Dynole 34-2 and siRNA against Dynamin 2 for blocking Dynamin-mediated processes in stimulated cells (Smith and Chircop, Traffic 2012).

Also, the assays to measure endocytosis are superficial and limited to the determination of the surface levels of various receptors.

Response: One of the principal function of Dynamins is to mediate transduction of ligand-bound receptors, which relies upon the internalization of these receptors leading to the downstream activation of signalling pathways. Therefore, we have measured the effect of Dynole 34-2 on both steps of this process, to ensure that the inhibition of multiple signalling pathways observed in the presence of Dynole 34-2 was resulting from the accumulation of multiple receptors at the surface of stimulated cells.

Finally, the authors argue that Dynamin 2 might be the main actor in this context, but the evidence in support of this contention is lacking.

Response: Using different cell lines and primary leukemias, we have shown that Dynole 34-2 prevents the transduction of multiple signalling pathways. Importantly, these signalling pathways are activated by 5 different ligands that bind to their own specific receptor, leading to the activation of different effectors following their internalization. The only shared process amongst all of those signalling pathways is Dynamin-mediated endocytosis, which we are targeting with Dynole 34-2.

REVIEWER COMMENTS

Reviewer #1 (Remarks to the Author):

In this revised version the authors have addressed a number of the concerns raised by the reviewers.

I have nevertheless still have some issues concerning the AML part.

They show that Dynole decrease in some AML patients not in all the CFC number. This assay is clearly not testing LSC nevertheless they mentioned that they reduced LSC. Please correct this statement. Also it is unclear what is the difference between patients that are responders versus non responders. A number of samples appears to behave similarly as normal CD34+ cells.

They then went on to test 2 AML who respond to Dynole ex vivo and injected them in vivo. In the results section they show a reduction in leukemic burden and an increase in survival especially when in combination. This data indicate that Dynole reduce leukemic burden in association with conventional chemotherapy but in discussion they state page 17 that they show the effect of Dynole on AML-LSC which is in reality not true as in order to prove an effect of ISC, they will have to show that they reduced the frequency of LSC in vivo by doing a limiting dilution analysis before and after treatment when leukemic come back.

Reviewer #2 (Remarks to the Author):

The authors answered to most of my major comments and added more results that further enrich the content of the manuscript. I have no major additional questions concerning their work. Congratulations to them all.

Reviewer #3 (Remarks to the Author):

Unfortunately, despite the effort to address criticisms by the reviewers#1 and 2, little has been done to provide definite proof that in the cell systems used Dynole acts specifically on Dynamin 2 as contented and by exclusively impairing endocytosis. We are full of examples of specific inhibitors that have additional targets besides the ones initially identified. A case in point is Dynasore, which acts specifically on Dynamin but depending on the dose and the cells in which it is used as additional inhibitory functions. This was shown by the De Camilli lab that used triple Dynamin fibroblasts to prove the point.

Here, work previously done on Dynole do argue about the fact that Dynamin2 activity is impaired, but to provide a definitive answer that what is seen here is exclusively due to Dynamin 2 inhibition, a siRNA, shRNA or CRISPR approach on at least one of the phenotypes seen appear mandatory. This could be easily done using BAFBa/F3 cell line expressing the receptor for IL-7, for example.

On endocytic assays, limiting the analysis to measuring the levels of cell surface is risky as additional effects and process can result in alteration of cell surface levels of receptors., as for instance altered recycling or defective delivery of biosynthetic cargos. Again using a more direct set of measurements in a dynamic and quantitative fashion for example in Ba/F3 cell line expressing the receptor for IL-7 might be sufficient.

The authors interpret their results of altered signalling assuming that all receptors would need to be internalized to signal appropriately, while this might be correct for some but not for all receptors. For example, inhibition of endocytosis might cause elevation of cell surface receptors and thus increases short term signalling, rather than inhibition of it.

Few follow up considerations:

-the authors cite few works as an indication of the specificity of Dynole 34-2; The first J. Med. Chem. 2009, 52, 12, 3762–3773 is the identification of the compound as an inhibitor of Dynamin 2 through medicinal chemistry approaches. Here the specificity is tested in vitro, while in vivo the authors used U2OS cells and showed inhibition of Transferrin at micromolar doses (the assays carried out is valid and was one of those requested to be done in their BA/F3 lines). Whether Dynole 34-2 has additional spurious effects or whether it is active or not in Dynamin 2 KD was not shown.

In the Gordon CP JMD 2013---the authors of the previous paper improved the medicinal chemistry approach to isolate Dynole 24-2 and not Dynole 34-2, used in the current study. Dynole 24-2 acts at low micromolar concentrations but the only cells tested in this work is again u2OS-fibrosarcoma. In the Smith and Chircop, Traffic 2012- HeLa cells are used along with a number of siRNA against canonical CME proteins that do inhibit clathrin-mediated endocytosis, similarly to Dynole 34-2 they also detect effects of endocytic proteins (and presumably dynole 34-2) independent on functional endocytosis: to quote these authors

...As endocytosis is inhibited at all mitotic stages in this study, this clearly shows that the mitotic roles of all CME proteins revealed in this study are independent of functional endocytosis, except at abscission.

Conclusions, while Dynole 34-2 inhibits dynamin 2 whether it does it in leukemic cells and its specificity in the relevant systems used has not been tested...

August 29, 2020

Dear reviewers,

Thank you for reviewing our manuscript entitled “Small molecule inhibition of Dynamin-dependent endocytosis targets multiple niche signals and impairs leukemia stem cells” for publication in *Nature Communications*. A point-by-point response to each of your comments is attached to this letter, with changes highlighted in the annotated and revised manuscript.

Yours sincerely,

Dr Cedric Tremblay

CONFLICT OF INTEREST: The authors declared no conflicts of interest.

Reviewers' comments:

Reviewer #1 (Remarks to the Author); expert in LSK and microenvironment:

In this revised version the authors have addressed a number of the concerns raised by the reviewers.

Response: First of all, we want to thank this reviewer for acknowledging our efforts to address all previous concerns expressed regarding the manuscript. This is very much appreciated.

I have nevertheless still have some issues concerning the AML part. They show that Dynole decrease in some AML patients not in all the CFC number. This assay is clearly not testing LSC nevertheless they mentioned that they reduced LSC. Please correct this statement.

Response: As suggested by this reviewer, we have removed any reference of direct targeting of LSCs in AML, and instead described the effect on colony numbers in the majority of primary AML samples tested.

Also it is unclear what is the difference between patients that are responders versus non responders. A number of samples appears to behave similarly as normal CD34+ cells.

Response: We agree with this reviewer, regarding the predictive biomarkers for efficacy of Dynole 34-2 in AML. However, this question is beyond the scope of the current study, which focuses on the pre-clinical efficacy of Dynole 34-2 as a novel combination therapy for acute leukemia. As observed with any given targeted therapy used for treating AML (Kotschy A, et al. Nature 2016 – figure 3), the wide range of response of primary AML samples reflect the heterogeneity of the disease itself. We have highlighted the wide range of responses to Dynole 34-2 with these primary AML samples in the revised version of the manuscript.

They then went on to test 2 AML who respond to Dynole ex vivo and injected them in vivo. In the results section they show a reduction in leukemic burden and an increase in survival especially when in combination. This data indicate that Dynole reduce leukemic burden in association with conventional chemotherapy but in discussion they state page 17 that they show the effect of Dynole on AML-LSC which is in reality not true as in order to prove an effect of LSC, they will have to show that they reduced the frequency of LSC in vivo by doing a limiting dilution analysis before and after treatment when leukemic come back.

Response: We agree with this reviewer comments that current *in vivo* experiments with AML PDX did not show direct effect of Dynole 34-2 on LSCs. Accordingly, we have removed any mention of the efficacy of Dynole 34-2 on LSCs in the biological context of AML. Instead we have rephrased the discussion to highlight the efficacy of Dynole 34-2 for killing leukemic cells that are responsible for relapse in human AML and T-ALL.

Reviewer #2 (Remarks to the Author); expert in T-ALL and mouse models:

The authors answered to most of my major comments and added more results that further enrich the content of the manuscript. I have no major additional questions concerning their work. Congratulations to them all.

Response: We thank this reviewer for the positive comments and for acknowledging our efforts to significantly improve the manuscript.

Reviewer #3 (Remarks to the Author); expert in endocytosis and signaling:

Unfortunately, despite the effort to address criticisms by the reviewers#1 and 2, little has been done to provide definite proof that in the cell systems used Dynole acts specifically on Dynamin 2 as contented and by exclusively impairing endocytosis. We are full of examples of specific inhibitors that have additional targets besides the ones initially identified. A case in point is Dynasore, which acts specifically on Dynamin but depending on the dose and the cells in which it is used as additional inhibitory functions. This was shown by the De Camilli lab that used triple Dynamin fibroblasts to prove the point.

Response: We note this reviewer's concerns, however our revisions were performed in accordance with the editorial comments provided via email, as follows: "Whilst we would expect you to respond to all of the reviewers' comments; in our editorial opinion, comments regarding normal haematopoiesis/HSPC, sensitivity of pre-LSC vs LSC (Reviewer #1 and #2), progression to leukaemia and mice survival (Reviewer #1) and relapse (Reviewer #2) are important points to experimentally address. Please highlight all changes in the manuscript text file. However, please bear in mind that we will be reluctant to approach the referees again in the absence of major revisions." We have already shown, as previously noted by this reviewer, that "Dynole 34-2 impairs Stat5 signaling and its inhibitory effect is relieved in Ba/F3 cells that express a constitutive active Stat5a", confirming that Dynole 34-2 kills cytokine-dependent leukemic cells by blocking signalling. We believe the experimental evidence for Dynole 34-2 mechanism of action in leukemic cells provided in **Fig. 1** and **S1** address the remaining point raised by this reviewer.

Here, work previously done on Dynole do argue about the fact that Dynamin2 activity is impaired, but to provide a definitive answer that what is seen here is exclusively due to Dynamin 2 inhibition, a siRNA, shRNA or CRISPR approach on at least one of the phenotypes seen appear mandatory. This could be easily done using Ba/F3 cell line expressing the receptor for IL-7, for example.

Response: As suggested by this reviewer, the use of a siRNA, shRNA or CRISPR approach for demonstrating the specificity of Dynole 34-2 for Dynamin 2 is technically impossible, given that our BaF3-IL7R cells require cytokine-induce signalling (and therefore, Dynamin-dependent endocytosis and signal transduction) for survival. Although we agree that the suggested experiments are conceptually crucial for demonstrating the specificity of Dynole 34-2, they simply cannot be performed in the biological context of cytokine-dependent leukemia. However, we have shown in Fig. 1g the effective killing of cytokine-dependent Ba/F3-IL7R cells by Dynole 34-2, with no efficacy in cytokine-independent Ba/F3-IL7R cells (overexpressing Stat5-CA), confirming that Dynole 34-2 kills cytokine-dependent leukemic cells by blocking signalling.

On endocytic assays, limiting the analysis to measuring the levels of cell surface is risky as additional effects and process can result in alteration of cell surface levels of receptors, as for instance altered recycling or defective delivery of biosynthetic cargos. Again, using a more direct set of measurements in a dynamic and quantitative fashion for example in Ba/F3 cell line expressing the receptor for IL-7 might be sufficient.

Response: We agree and therefore, we measured the dynamic changes in colocalization of IL-7R with clathrin and the early-endosome marker Rab5 in Ba/F3-ILR cells, as previously described (Tremblay CS, et al. Leukemia 2016; Henriques CM, et al. Blood 2010). Our results demonstrate that Dynole 34-2 prevents the internalization of IL-7R cells in leukemic cells, as measured by the decreased distribution of IL-7R in Rab5-positive early endosomes following IL-7 stimulation (**Fig. 1d-e** and **S1d**). These dynamic and quantitative assays provide direct evidence of the inhibitory effect of Dynole 34-2 on signalling receptors in leukemic cells, which addresses the point raised by this reviewer.

The authors interpret their results of altered signalling assuming that all receptors would need to be internalized to signal appropriately, while this might be correct for some but not for all receptors.

Response: We thank this reviewer for raising this critical point, and we performed dynamic studies to show that Dynole 34-2 inhibited signal transduction by preventing IL-7R internalization in Ba/F3-IL7R cells (**Fig. 1d-e** and **S1d**). Although we agree with this reviewer that not all receptors need to be internalized to signal appropriately, the receptors tested in the current manuscript require internalization for triggering downstream signalling, as summarized in Cendrowski J, et al. Cytokine & Growth Factor Reviews 2016 and previously described in Chapman G, et al. Biochimica et Biophysica Acta 2016 for Notch1.

For example, inhibition of endocytosis might cause elevation of cell surface receptors and thus increases short term signalling, rather than inhibition of it.

Response: Although we agree that Dynamin inhibition has the potential to increase receptor signalling, our *in vitro* studies in leukemic cell lines and primary leukemias support our claim that inhibition of DDE by Dynole 34-2 prevents receptor internalization and downstream activation of signalling pathways (Fig. **1b**, **2b-c** and **S8d**). Moreover, we have performed dynamin colocalization assays confirming that Dynole 34-2 prevented the distribution of IL-7R into Rab5-positive early endosomes following IL-7 stimulation (**Fig. 1d-e** and **S1d**), and showed that dose-dependent accumulation of IL-7R the surface of Ba/F3-IL7R cells correlated with decreased survival. Altogether, these data suggest that Dynole 34-2 kills cytokine-dependent leukemic cells by inhibiting DDE of signalling receptors.

Few follow up considerations:

-the authors cite few works as an indication of the specificity of Dynole 34-2; The first J. Med. Chem. 2009, 52, 12, 3762–3773 is the identification of the compound as an inhibitor of Dynamin 2 through medicinal chemistry approaches. Here the specificity is tested *in vitro*, while *in vivo* the authors used U2OS cells and showed inhibition of Transferrin at micromolar doses (the assays carried out is valid and was one of those requested to be done in their BA/F3 lines). Whether Dynole 34-2 has additional spurious effects or whether it is active or not in Dynamin 2 KD was not shown.

In the Gordon CP JMD 2013---the authors of the previous paper improved the medicinal chemistry approach to isolate Dynole 24-2 and not Dynole 34-2, used in the current study. Dynole 24-2 acts at low micromolar concentrations but the only cells tested in this work is again U2OS-fibrosarcoma.

In the Smith and Chircop, Traffic 2012- HeLa cells are used along with a number of siRNA against canonical CME proteins that do inhibit clathrin-mediated endocytosis, similarly to Dynole 34-2 they also detect effects of endocytic proteins (and presumably dynole 34-2) independent on functional endocytosis: to quote these authors... As endocytosis is inhibited at all mitotic stages in this study, this clearly shows that the mitotic roles of all CME proteins revealed in this study are independent of functional endocytosis, except at abscission.

Conclusions, while Dynole 34-2 inhibits dynamin 2 whether it does it in leukemic cells and its specificity in the relevant systems used has not been tested...

Response: We appreciate the constructive comments of this reviewer. In contrast to previous work performed with cytokine-independent cancer cell lines (Smith and Chircop, Traffic 2012; Chircop M, et al. Mol Cancer Ther 2011), our current study is using Dynole 34-2 to target multiple signalling pathways in cytokine-dependent leukemia cells. Using Ba/F3-IL7R cells, we have shown that: 1) Dynole 34-2 impairs Stat5 signalling and its inhibitory effect is relieved in cells that express a constitutive active Stat5 – confirming that Dynole 34-2 kills cytokine-dependent leukemic cells by blocking signalling; 2) dose-dependent effect of Dynole 34-2 on pStat5 levels correlated with survival; 3) dose-dependent accumulation of IL-7R induced by Dynole 34-2 correlated with apoptosis. Although these experiments do not address the specificity of Dynole 34-2 as a Dynamin inhibitor, they clearly establish that Dynole 34-2 kills cytokine-dependent leukemic cells by blocking signal transduction.

REVIEWERS' COMMENTS

Reviewer #1 (Remarks to the Author):

In this second revision, the authors have addressed in my view the major concerns raised by the reviewers. I believe this new version present a major improvement of the message whichh will be of interest for scientists working in the field.

Reviewer #3 (Remarks to the Author):

The manuscript is significantly improved.

The finding that Dynole 34-2 kills cytokine-dependent leukemic cells by blocking signal transduction is indeed well established and demonstrated. The specificity of this compound in mediating solely Dynamin dependent process remains unproven. The authors argue that BaF3-IL7R requires IL73R signaling for survival. However, it should be possible to perform transient and acute siRNA experiments to assess the impact of specifically eliminating Dynamin 2.

In the absence of these experiments, the authors should at least explicitly state that additional molecular genetics experiments would be needed to establish unequivocally the mechanisms of action of Dynole 34-2 as actin exclusively on dynamin.

October 7, 2020

Dear reviewers,

Thank you for providing constructive input that contributed to improve our manuscript entitled “Small molecule inhibition of Dynamin-dependent endocytosis targets multiple niche signals and impairs leukemia stem cells” as part of the submission process for publication in *Nature Communications*. A point-by-point response to each of your comments is attached to this letter, with changes highlighted in the annotated and revised manuscript.

Yours sincerely,

Dr Cedric Tremblay

CONFLICT OF INTEREST: The authors declared no conflicts of interest.

Reviewers' comments:

Reviewer #1 (Remarks to the Author); expert in LSK and microenvironment:

In this second revision, the authors have addressed in my view the major concerns raised by the reviewers. I believe this new version present a major improvement of the message which will be of interest for scientists working in the field.

Reviewer #3 (Remarks to the Author); expert in endocytosis and signaling:

The manuscript is significantly improved.

Response: We thank this reviewer for the positive comments and for acknowledging our efforts to significantly improve the manuscript.

The finding that Dynole 34-2 kills cytokine-dependent leukemic cells by blocking signal transduction is indeed well established and demonstrated. The specificity of this compound in mediating solely Dynamin dependent process remains unproven. The authors argue that BaF3-IL7R requires IL7R signaling for survival. However, it should be possible to perform transient and acute siRNA experiments to assess the impact of specifically eliminating Dynamin 2.

In the absence of these experiments, the authors should at least explicitly state that additional molecular genetics experiments would be needed to establish unequivocally the mechanisms of action of Dynole 34-2 as acting exclusively on dynamin.

Response: We note this reviewer's concerns regarding the specificity of Dynole 34-2 in the biological context of leukemia, and we agree that further molecular genetic assays are required to demonstrate that Dynole 34-2 prevents the internalization of signalling receptors by specifically inhibiting Dynamin 2 in leukemic cells. Therefore, as suggested by this reviewer, we have included the following sentence in the Discussion section: Our data support previous reports confirming the efficacy of Dynole 34-2 to inhibit internalization of signalling receptors and downstream signal transduction in different malignancies (Chew HY et al., *Cell* 2020; Luwor R et al., *Cancer Investigation* 2019). However, further genetic molecular assays with cytokine-dependent cells lacking Dynamin or expressing specific mutants of Dynamin will be required to characterize the specificity of Dynole 34-2 for targeting Dynamin in the biological context of leukemia.